# Nucleosome wrapping states encode principles of 3D genome organization

Zengqi Wen [1] ✉, Ruixin Fang[2], Ruxin Zhang[2], Xinqian Yu[2], Fanli Zhou[1] & Haizhen Long [2] ✉

Nucleosome is the basic structural unit of the genome. During processes like DNA replication and gene transcription, the conformation of nucleosomes undergoes dynamic changes, including DNA unwrapping and rewrapping, as well as histone disassembly and assembly. However, the wrapping characteristics of nucleosomes across the entire genome, including region-specificity and their correlation with higher-order chromatin organization, remains to be studied. In this study, we investigate the wrapping length of DNA on nucleosomes across the whole genome using wrapping-seq. We discover that the chromatin of mouse ES cells forms Nucleosome Wrapping Domains (NRDs), which can also be observed in yeast and fly genomes. We find that the degree of nucleosome wrapping decreases after DNA replication and is promoted by transcription. Furthermore, we observe that nucleosome wrapping domains delineate Hi-C compartments and replication timing domains. In conclusion, we have unveiled a previously unrecognized domainization principle of the chromatin, encoded by nucleosome wrapping states.

In eukaryotic organisms, genomic DNA wraps with histones to form chromatin. The nucleosome is the fundamental structural unit of chromatin. The first high-resolution (2.8 Å) crystal structure of the nucleosome was determined by Luger et al.[1]. The structure revealed that a nucleosome core particle contains 147 base pairs of DNA wrapping about 1.65 turns around the surface of histone octamer. Multiple interactions are formed within the histone octamer and between the histone octamer and DNA, thus forming a stable complex. Nucleosome is assembled in a stepwise fashion[2]. An H3-H4 tetramer initially associates with DNA to form a tetrasome, which can then be followed by the addition of one copy of H2A-H2B to form a hexasome. Alternatively, two copies of H2A-H2B can be consecutively added to create a complete nucleosome. Thus, intermediate states, including DNA unwrapping and octamer dissociation, among others may exist during DNA replication when nucleosomes are disassembled ahead of replication fork and re-assembled behind the replication fork.

As a large complex, nucleosome is dynamic in nature. For instance, DNA located on nucleosomes can rapidly transit between "unwrapping" and "rewrapping" states, a phenomenon referred to as "DNA breathing". Utilizing fluorescence resonance energy transfer (FRET), it was demonstrated that DNA on nucleosomes spends approximately 2–10% of its time in the unwrapped state[3,4]. Significantly, this characteristic is also observed in nucleosomes within nucleosome arrays[5]. Moreover, the dynamics of nucleosomes within the cell are regulated during various biological processes. Chromatin remodeling complexes are able to regulate the structure of nucleosomes using the energy generated by ATP hydrolysis[6]. For instance, the nucleosome remodeling complex INO80, can disrupt interactions between H2A and DNA, leading to the opening of approximately 15 base pairs of DNA on the nucleosome[7,8]. Histone chaperone complexes can bind to histones and regulate the assembly or disassembly of nucleosomes. For instance, the SPT16 subunit of the FACT complex can displace H2A-H2B dimers[9,10], thus facilitating the formation of an open nucleosome structure. In the context of gene transcription, RNA polymerase can induce the opening of DNA on the nucleosome, leading to the formation of unwrapped nucleosomes. This process produces intermediate states such as unwrapping DNA of 20 base pairs, 50 base pairs, 60 base pairs, and so forth[10,11].

[1]School of Medicine, Shenzhen Campus of Sun Yat-Sen University, Sun Yat-Sen University, Shenzhen, Guangdong 518107, China. [2]Institute of Molecular Physiology, Shenzhen Bay Laboratory, Shenzhen 518132, China. ✉e-mail: wenzq7@mail.sysu.edu.cn; longhaizhen@szbl.ac.cn

While various nucleosome wrapping states have been observed under in vitro conditions, the genuine nucleosome wrapping states within cells remain significantly less characterized. By employing ChIP-exo, Rhee et al. found that histones are asymmetrically depleted on +1 nucleosomes in yeast, leading to the formation of subnucleosomes[12]. Another widely used approach for mapping nucleosome states is MNase-seq, which directly measures the length of DNA fragments protected by nucleosomes. A recent study from Henikoff's lab suggested that the protected short DNA fragments associated with nucleosomes can be indicative of subnucleosomal states[13]. In our previous study, we employed MNase-X-ChIP-seq to map the nucleosome wrapping states in mouse ES cells[14]. We found that H2A.Z nucleosomes are more unwrapped than canonical nucleosomes, and the wrapping states of H2A.Z nucleosomes is correlated with gene transcription activity[14].

As those studies of mapping nucleosome states within cells have primarily focused on analyzing the wrapping states of the +1 nucleosomes, we aim to address the characteristics of nucleosome wrapping states across the entire genome in this study. To this end, we have adapted from our previous MNase-X-ChIP-seq protocol[14] and utilized the lengths of DNA fragments generated by MNase enzyme cleavage to calculate quantitative metrics of nucleosome wrapping, namely Nucleosome Wrapping Score (NRS) and Nucleosome Wrapping Index (NRI), which is the z-score transformation of NRS. These two metrics were used to characterize the average wrapping states of nucleosomes within a certain genomic interval. To distinguish this experimental and analytical workflow from classic MNase-seq and emphasize the analysis of nucleosome wrapping states, we propose referring to the pipeline as "wrapping-seq". Accordingly, H3-wrapping-seq and xMNase-wrapping-seq refer to nucleosome wrapping analysis using H3 MNase-X-ChIP-seq data and crosslinked MNase-seq data, respectively.

In summary, we revealed that the genomic chromatin of mouse ES cells forms Nucleosome Wrapping Domains (NRDs), including tightly wrapped NRDs (TiNRDs) and loosely wrapped NRDs (LoNRDs). Formation of NRDs is conserved in yeast and fly genomes. TiNRDs and LoNRDs precisely coincide with Hi-C A compartment domains and B compartment domains, respectively. Interestingly, our data suggests that nucleosomes in euchromatin chromatin (e.g. A compartments) exhibit tighter wrapping compared to nucleosomes in heterochromatin (e.g. B compartments). Furthermore, we showed that transcription enhances nucleosome wrapping in genic regions.

## Results

### Nucleosome wrapping index is robust for nucleosome wrapping state characterization

To study the wrapping states of nucleosomes across genome-wide, we performed H3-wrapping-seq and sequenced the library deeply. In total, 210 M high quality paired-end reads were generated after filtering. Then we use fragments within 50–250 bp to analyze genome-wide nucleosome wrapping states (Supplementary Fig. S1a). As shown in Fig. 1a, to calculate the NRS (nucleosome wrapping score) for a genomic interval (e.g., a 10 Kb bin), we first counted the length of all the DNA fragments mapped within that 10 Kb interval. Then we separated the DNA fragments into two groups by a fragment length "break point (bkp)" $X$ bp. NRS is computed as the relative deviation between the number of fragments within $X$–250 bp (represented by variable "a") and the number of fragments within 50-$X$ bp (represented by variable "b"), expressed as $NRS(X) = (a - b)/(a + b)$. For example, if the "break point" is 140 bp, then NRS(140) is computed as the relative deviation between the number of DNA fragments within 140–250 bp (longer ones) and the number of DNA fragments within 50–140 bp (shorter ones). Thus, larger NRS value means the nucleosomes provided more protection for the nucleosomal DNA during MNase digestion, indicating DNA may wrap tighter on nucleosomes; vice versa, smaller NRS value indicates DNA may wrap looser on

nucleosomes. It's reasonable that when the break point increased from 80 bp to 160 bp with 10 bp step, the genome-wide NRS generally shifted from 1 to −1 (Fig. 1b, Supplementary Fig. S1b). However, when the raw NRS of each break point was transformed as z-score to derive NRI (nucleosome wrapping index), the genome-wide NRIs of different break points show highly similar pattern (Fig. 1c, Supplementary Fig. S1c) and high correlation (Supplementary Fig. S1d). Thus, z-score transformation eliminates the bias of nucleosome wrapping degree introduced by selecting the break point arbitrarily. Similarly, larger NRI value indicates DNA may wrap tighter on nucleosomes; smaller NRI value indicates DNA may wrap looser on nucleosomes. While NRS is an absolute metric of wrapping degree, NRI is a relative metric of wrapping degree, representing the ordering of nucleosome wrapping degree of all the genomic intervals. We will use break point of 140 bp to calculate NRS and NRI in the following text unless indicated.

Next, we computed NRS and NRI of H3 with bin size of 1 Kb, 5 Kb, 10 Kb, 50 Kb and 100 Kb (genomic intervals of 1 Kb, 5 Kb, 10 Kb, 50 Kb and 100 Kb). While NRSs show variation with different bin sizes (Supplementary Fig. S1e), the patterns of NRIs are rather constant, and highly correlated at genome-wide (Supplementary Fig. S1e, f). The bin size of 100 kb will be used for most downstream analysis unless noted. When NRI(140) was calculated for histone H4 wrapping-seq data and the input data of MNase-X-ChIP, they both show high correlation with H3 NRI(140) (Fig. 1d, Supplementary Fig. S1g), suggesting that MNase-seq data can be used to analyze nucleosome wrapping states.

To further test whether MNase NRI is sensitive to variation of MNase digestion, we prepared xMNase-wrapping-seq libraries within the digestion time of 10 - 50 min, when mono-nucleosomal DNA was more than other single DNA bands, such as di- or tri- nucleosomal DNA (Supplementary Fig. S1h). We found that the genome-wide MNase NRS(140) generally shift from positive value to negative value (Fig. 1e, Supplementary Fig. S1i), as a result of decreased proportion of longer DNA fragments and increased proportion of shorter DNA fragments when the digestion time increases. However, the NRI(140) patterns from different digestion time points are highly similar (Fig. 1e, Supplementary Fig. S1j), and highly correlated (Supplementary Fig. S1k), suggesting that MNase NRI is not sensitive to MNase digestion variation, as long as mono-nucleosome is the dominant over other single DNA band.

We observed that the nucleosome density represented by H3 or H4 MNase-X-ChIP signals or MNase-X-ChIP input signal show obvious variation across the genome (Fig. 1d). Surprisingly, we observed that NRI is highly positively correlated with nucleosome density (Fig. 1d–f). As higher nucleosome density might provide more protection of MNase digestion, resulting in higher NRI value, we intended to test whether the NRI pattern still persist when there should be no variation of nucleosome density across the genome. However, before that, we also need to figure out whether the variation of nucleosome density is due to potential extraction bias of chromatin or DNA during wrapping-seq or due to genome copy number variation as a result of DNA replication in S-phase cells. To this end, we sorted cells by G1, S and G2/M phases, and then performed xMNase-wrapping-seq using cross-linked chromatin, and genome sequencing using genomic DNA ("gDNA") in parallel. We found that in all three cell cycle phases, the signals of genomic DNA and MNase digested chromatin are highly similar viewed from IGV (Fig. 1g) and highly correlated at genome-wide (Supplementary Fig. S1l). As the genomic DNA libraries represent genuine genome coverage, these results suggested that the wrapping-seq procedure did not introduce bias of nucleosome coverage of the genome. Moreover, we found that both chromatin and genomic DNA signals show more variation along the genome of S-phase cells than that of G1 or G2/M phase cells (Fig. 1g). To quantify the variation, we counted the genomic DNA and chromatin signals in 100 Kb genomic intervals, and showed the distribution of signal as histograms. As shown in Fig. 1h, i, S phase cells show a bimodal distribution of both

genomic DNA and chromatin signals, whereas G1 and G2/M cells show a unimodal distribution of those signals. Moreover, the peak width of S phase cell signals is wider than that of G1 and G2/M cells. These results supported that the genomic DNA and chromatin signals show more variation in S phase cells than in G1 and G2/M cells. However, when we calculated the NRI pattern using the chromatin signal, we found that the NRI patterns of G1, S and G2/M phase cells are highly similar (Fig. 1g), and are equally well correlated with the MNase NRI of

unsynchronized cells (Supplementary Fig. S1m). Thus, these results argue that the NRI pattern is not an artificial consequence of nucleosome density variation along the genome.

Taken together, these results demonstrate that quantification of nucleosome wrapping states by nucleosome wrapping index is highly robust against histone types, genomic resolution and variation of MNase digestion. Notably, this method is also very straightforward when taking advantage of MNase-seq data.

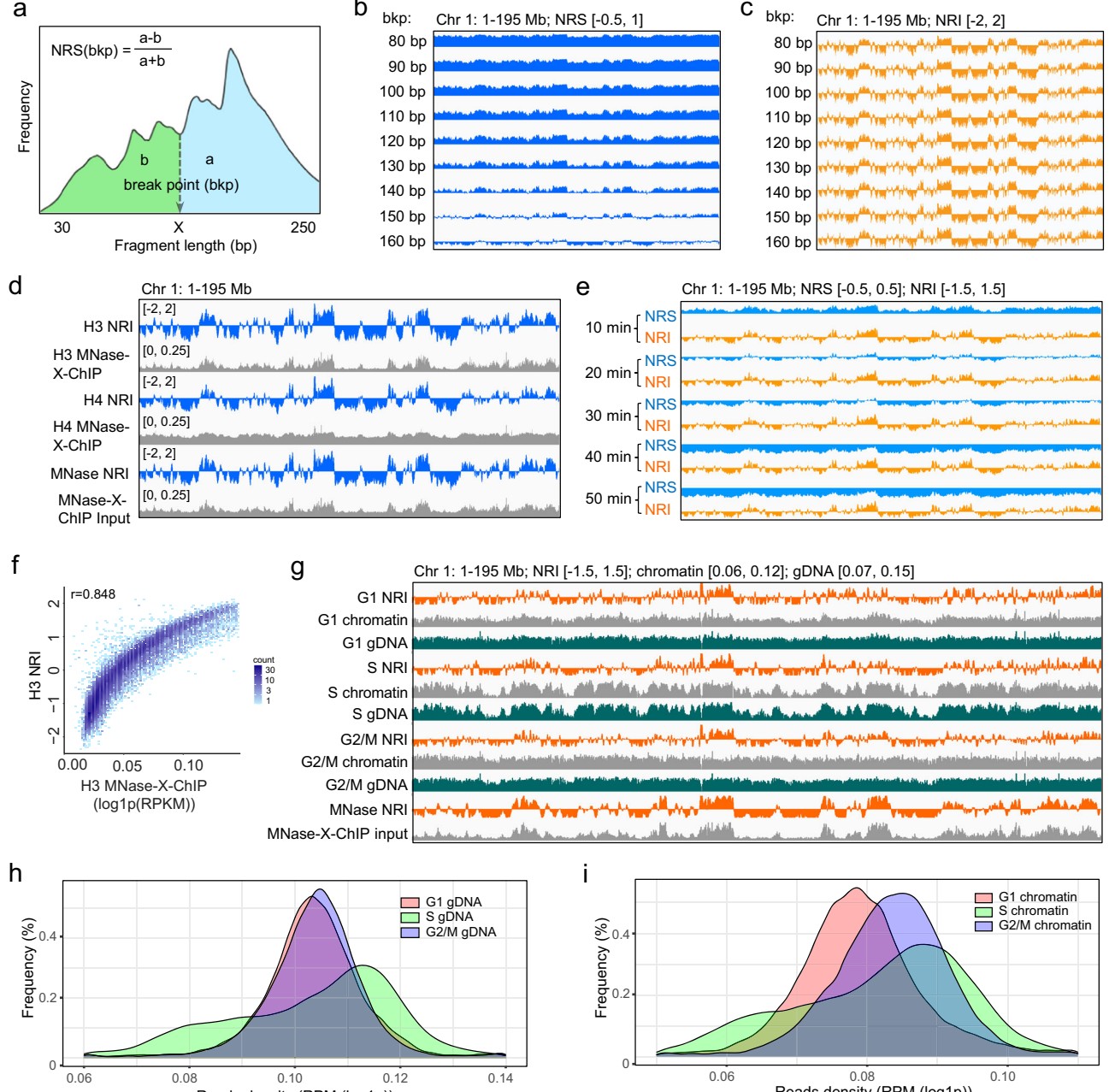

**Fig. 1 | Nucleosome wrapping index is robust for nucleosome wrapping state characterization. a** Diagram shows the calculation of NRS, "a" is the number of fragments within $X$–250 bp and "b" is the number of fragments within 50-$X$ bp. b-c, IGV tracks show NRS (**b**) or NRI (**c**) of chromosome 1 when different break point is chosen, including 80 bp, 90 bp, 100 bp, 110 bp, 120 bp, 130 bp, 140 bp, 150 bp, 160 bp. **d** IGV tracks show the similarity between NRIs of H3 ChIP, H4 ChIP or MNase-X-ChIP input on Chromosome 1. **e** IGV tracks show the dynamic changes of NRSs and NRIs under time-course MNase digestion on Chromosome 1. **f** Dot plot shows the positive correlation between H3 NRI and H3 genome coverage counted with 100 Kb bins ($n = 27,268$). "r" indicates the Pearson correlation coefficient. **g** IGV tracks show NRIs of xMNase-seq, MNase digested chromatin signals and genomic DNA signals of G1, S and G2/M cells on Chromosome 1. Histograms show the reads density of genomic DNA (**h**) or MNase digested chromatin (**i**) of G1, S and G2/M cells. The signal ranges are indicated at the top of figures for panels **b**, **c**, **e**, **g**. Source data are provided as a Source Data file.

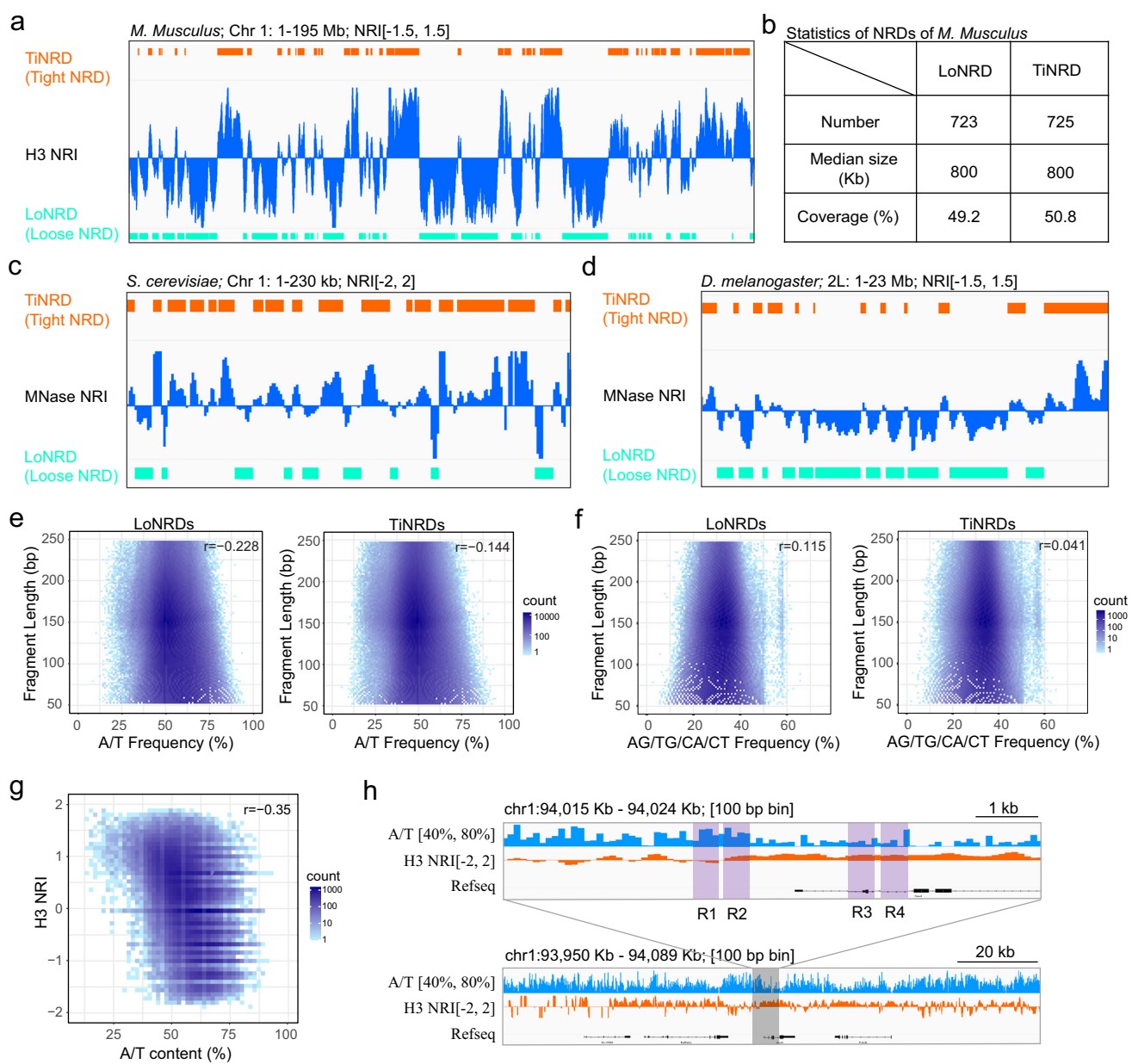

**Fig. 2 | Nucleosome wrapping domain is detected in yeast, fly and mouse genome. a** IGV tracks show the distribution of TiNRDs (tight nucleosome wrapping domains) and LoNRDs (loose nucleosome wrapping domains) on Chromosome 1. **b** Table shows the number, median length and genome coverage of TiNRDs and LoNRDs of mouse genome. IGV tracks show the distribution of TiNRDs and LoNRDs on Chromosome 1 of *S. cerevisiae* genome (**c**) or chromosome 2 L of *D. melanogaster* genome (**d**). Dot plots show the correlation between A/T content of a DNA fragment and its' length (**e**), or between AG/TG/CA/CT dinucleotide frequency of a DNA fragment and its' length (**f**) within LoNRDs (*n* = 723) and TiNRDs (*n* = 725). **g** Dot plot shows the correlation between genome A/T content and H3 NRI counted with sampled 100 bp bins (*n* = 200000). "r" indicates the Pearson correlation coefficient. **h** IGV tracks genome-wide distribution of A/T content and H3 NRI counted with 100 bp bin size. Regions "R1" and "R2" have comparable A/T content, but H3 NRI varies greatly; Regions "R3" and "R4" have apparently different A/T content, but the H3 NRI values are similar. The signal ranges are indicated at the top of figures for panels **a**, **c**, **d**. Source data are provided as a Source Data file.

## Nucleosome wrapping domain is detected in yeast, fly and mouse genome

Interestingly, we observed from the genome-wide NRI pattern that there are large genome domains continuous with characteristic nucleosome wrapping states (Fig. 1c, d, e, g). To qualify the nucleosome wrapping states along the genome, we segmented the genome based on NRI(140) of histone H3 (Fig. 2a). In total, the genome is resolved into 1448 Nucleosome Wrapping Domains (NRDs) (Fig. 2b). NRDs enriched with positive NRI were termed as tight NRDs (TiNRDs, *n* = 723); NRDs enriched with negative NRI were termed as loose NRDs (LoNRDs, *n* = 725) (Fig. 2a, b). Both TiNRDs and LoNRDs in mouse ES

cells have a median length of 800 Kb, and cover 49.2% and 50.8% of mouse genome, respectively (Fig. 2b). These NRDs maintain characteristic nucleosome wrapping features across multiple NRI groups (Supplementary Fig. S2a), bin resolution (Supplementary Fig. S2b) and histone types (Supplementary Fig. S2c), suggesting that the qualification of nucleosome wrapping states by NRDs (refers to NRDs of H3 NRI(140) unless stated) is robust.

To explore whether the domainization of nucleosome wrapping is conserved in species other than *M. musculus*, we calculated NRI using MNase-seq data of *S. cerevisiae*[15] and *D. melanogaster* S2 cells[16], respectively. As in mouse ES cells, we observed stereotypical

nucleosome wrapping domains in both *S. cerevisiae* (Fig. 2c, Supplementary Fig. S2d) and *D. melanogaster* (Fig. 2d, Supplementary Fig. S2e). Thus, these results suggest that the domainization of nucleosome wrapping is conserved during evolution.

It's reported that MNase has a sequence bias towards A/T[17,18]. Consistently, when analyzing the DNA fragments of H3 wrapping-seq, we observed high A/T frequency at both upstream and downstream of the cutting sites of DNA fragments, mapped within either TiNRDs or LoNRDs (Supplementary Fig. S2f). We also observed high G frequency downstream of A/T or high C frequency upstream of A/T, at the direction of top strand (Supplementary Fig. S2f). Although the DNA fragments from TiNRDs have lower average A/T content than those from LoNRDs, they show no substantial difference in MNase cutting bias, suggesting that the sequence bias of MNase is not systematically biased towards TiNRDs or LoNRDs. As each DNA fragments from H3 wrapping-seq represents a single nucleosome cut out from the chromatin, we analyzed the correlation between fragment length and the A/T content or dinucleotide frequency of A/TG and CA/T for each DNA fragment. We found that the correlations are weak for DNA fragments from either TiNRDs or LoNRDs (Fig. 2e, f), suggesting that the A/T content or dinucleotide frequency per se do not determine the DNA length protected by a nucleosome. To analyze the genome-wide correlation between the A/T content and H3 NRI at single nucleosome-level, we used 100 bp bin size to calculate and count the signals of A/T content and H3 NRI. We found that the Pearson correlation coefficient between A/T content and H3 NRI is −0.354 (Fig. 2g), indicating that the genome-wide A/T content may contribute, but mildly to the regulation of nucleosome wrapping states. As shown by IGV tracks, for example, regions "R1" and "R2" have comparable A/T content, but H3 NRI varies greatly; regions "R3" and "R4" have apparently different A/T content, but the H3 NRI values are similar (Fig. 2h). Taken together, we did observe A/T bias of MNase during chromatin digestion, but we did not observe a strong correlation between genome A/T content and DNA fragment length or nucleosome wrapping index at single nucleosome level.

### Nucleosomes wrap tighter in euchromatin than in heterochromatin

To investigate the relationship between nucleosome wrapping and chromatin states, we computed the genome-wide correlation between H3 NRI(140) and DNase I sensitivity, H3K4me1, H3K4me3, H3K9me3 or H3K27me3. The signals of histone modifications were normalized by histone signal in consideration of the variation of nucleosome density. We found that H3 NRI(140) is well positively correlated with DNase I sensitivity, and active chromatin markers H3K4me1 and H3K4me3, but not with heterochromatin markers H3K9me3 or H3K27me3 (Fig. 3a). We further counted the fragment length in the peak regions of H3K4me1, H3K4me3, H3K9me3 and H3K27me3. We found that euchromatin regions with H3K4me1 or H3K4me3 peaks show higher proportion of long DNA fragments than heterochromatin regions with H3K9me3 or H3K27me3 peaks (Fig. 3b), and that heterochromatin regions show higher proportion of short DNA fragments than euchromatin regions (Fig. 3b). Correspondingly, quantification of NRS in these peak regions showed that nucleosomes in H3K4me1 or H3K4me3 peaks regions have higher H3 NRS(140) than those in H3K9me3 or H3K27me3 peaks regions (Supplementary Fig. S3a). On the other hand, when analyzing the epigenetic features of NRDs, we found that TiNRDs show higher level of euchromatin marks, such as H3K4me1, H3K4me3 than LoNRDs (Supplementary Fig. S3b); while LoNRDs show higher level of heterochromatin marks such as H3K9me3 and H3K27me3 than TiNRDs (Supplementary Fig. S3b). Together, these results suggested that nucleosomes may wrap tighter in euchromatin than in heterochromatin.

To explore the relationship between nucleosome wrapping and transcription activity, we computed the correlation between NRI and

Polr2a ChIP signal or total RNA-seq signal at gene bodies. We found that NRI is highly correlated with Polr2a ChIP signal, but to a weaker extend with total RNA level (Fig. 3c), suggesting that NRI is closely related to the RNA polymerase activity on the genes. After sorting the gene bodies according to Polr2a ChIP signal, we found that NRI decreased gradually along RNAPII signal (Fig. 3d, Supplementary Fig. S3c). Moreover, NRI is highly enriched immediately downstream of TSS, and decreased within the gene bodies (Fig. 3d, e), suggesting +1 nucleosome is highly wrapped. This result is consistent with that +1 nucleosome is involved in the regulation of RNA Polymerase initiation[19,20]. At the transcription termination sites (TTSs), we found that NRI is low at the center of TTS, but it increases downstream beyond the TTS (Fig. 3f), suggesting there are other mechanisms regulating nucleosome wrapping beyond transcription elongation.

### Transcription promotes nascent nucleosome wrapping

During DNA replication, the nucleosomes are dis-assembled ahead of the DNA replication fork, and re-assembled behind the replication fork. To study the nucleosome wrapping dynamics during DNA replication, we pulse-labeled mouse ES cells with EdU to label nascent nucleosomes (referred to as "Pulse" condition), then chased the cells with thymidine to follow the maturation dynamics of nascent nucleosomes (referred to as "Chase" condition) (Fig. 4a). As only a small portion of genome for each S-phase cell is labeled under the labeling condition, the "Input" sample contains majorly parental nucleosomes with steady-state of wrapping, while the "Pulldown" sample contains nascent nucleosomes. To test whether EdU labeling affects nucleosome wrapping, we extensively labeled the mouse ES cells with EdU for 16 h, then performed xMNase-wrapping-seq. We found that the genome-wide NRIs of labeled and non-labeled cells are the same (Supplementary Fig. S4a), and highly correlated at genome-wide (Supplementary Fig. S4b). These results suggest that EdU labeling does not have a significant effect on nucleosome wrapping states. Then we analyzed the nucleosome wrapping states of both parental and nascent nucleosomes. We found that the genome-wide NRI is highly correlated and stable between the parental nucleosomes and nascent nucleosomes at genome-wide (Supplementary Fig. S4c) and in NRDs (Supplementary Fig. S4d), under both "Pulse" and "Chase" conditions, suggesting that the NRDs are stably inherited during DNA replication.

Next, we used the NRS metric to compare the wrapping dynamics of nascent nucleosomes, as explained below. In either "Pulse" or "Chase" condition, as the parental nucleosomes and nascent nucleosomes are within the same cell, they underwent exactly the same MNase digestion. Thus, the NRSs of parental nucleosomes and nascent nucleosomes can be compared directly within "Pulse" or "Chase" condition. Moreover, as the parental nucleosomes are at steady-state, they should have the same NRS between "Pulse" and "Chase" conditions. Thus, we used the NRS of parental nucleosomes as a "baseline" to analyze the dynamics of nascent nucleosomes. Interestingly, we observed that while the NRS of nascent nucleosomes is smaller than that of parental nucleosome under "Pulse" condition, it became higher than that of parental nucleosomes under "Chase" condition, in both TiNRDs and LoNRDs (Fig. 4b). This result suggested that, after DNA replication, the DNA wraps looser on the nascent nucleosomes than on the parental nucleosomes. However, within a time window, the nascent nucleosomes will mature and the DNA will wrap tighter than parental nucleosomes. This dynamic wrapping states of nascent nucleosomes can also be observed in the MINCE dataset of Drosophila S2 cells[16] (Supplementary Fig. S4e). Together, these results suggested that the wrapping state of nascent nucleosomes is gradually matured after DNA replication, and this process is independent of the local nucleosome wrapping domains.

To explore whether transcription regulates nucleosome wrapping, we inhibit transcription with triptolide (TPL)[21] during DNA

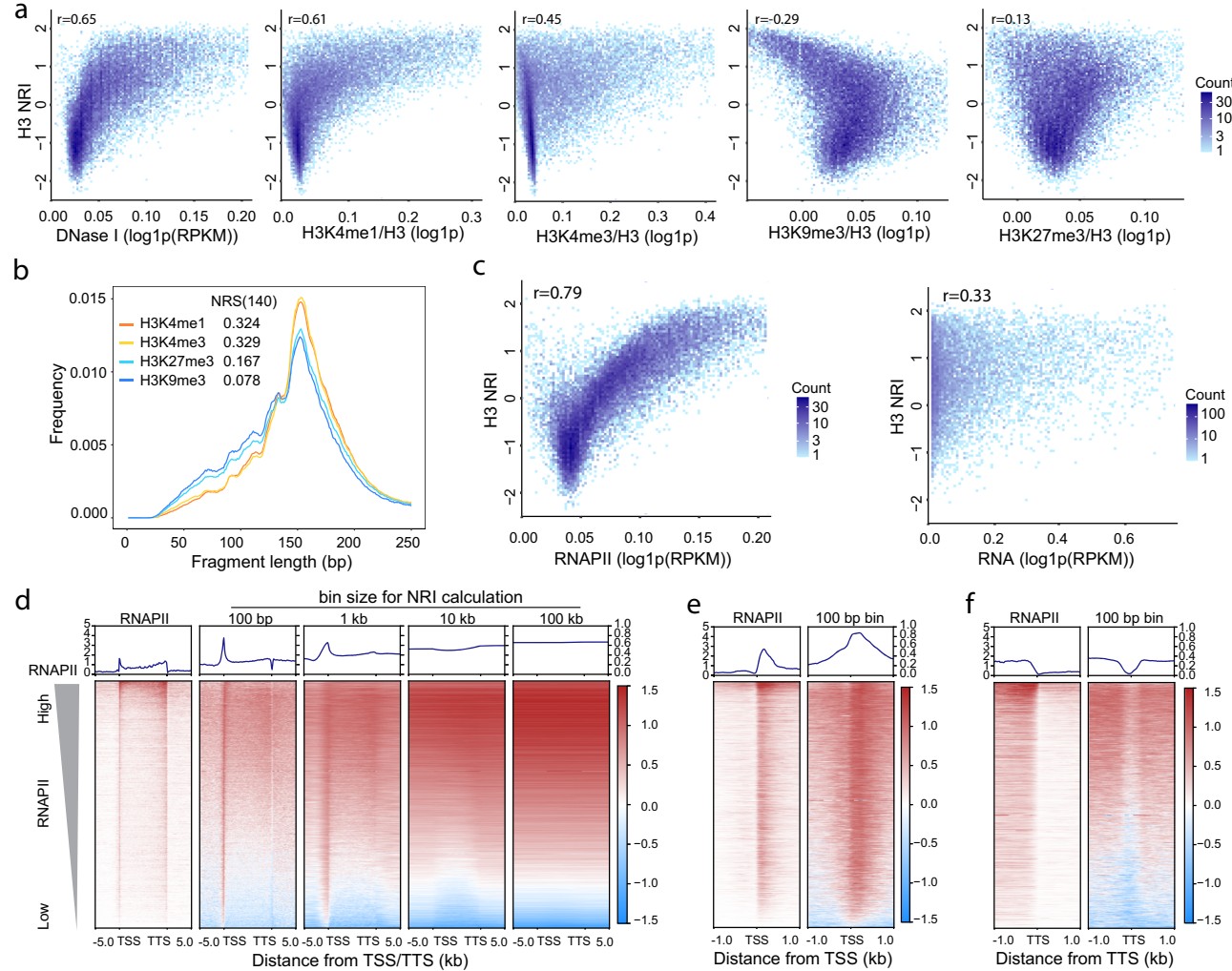

**Fig. 3 | Nucleosomes wrap tighter in euchromatin than in heterochromatin.**
**a** Dot plots show the genome-wide correlations between NRI(140) of H3 and DNase
I, H3K4me1, H3K4me3, H3K9me3 or H3K27me3 counted with 100 Kb bins
($n = 27,268$). The signal of histone modifications was normalized by H3.
**b** Histogram shows the distribution of fragment length of histone H3 enriched DNA
fragments in the peak regions of H3K4me1, H3K4me3, H3K9me3 and H3K27me3.
NRS(140)s are the nucleosome wrapping score calculated within the peaks of each
markers. **c** Dot plots show the correlations between NRI(140) of H3 and Polr2a or

total RNA-seq signal within gene body regions ($n = 43511$). **d**–**f** Heatmaps show the
distribution of H3 NRI at gene body regions (**d**), 2 kb regions around transcription
start sites (TSSs) (**e**) and 2 kb regions around transcription termination sites (TTSs)
(**f**). Genes were sorted in descending order according to Polr2a signal in the gene
body regions ($n = 43511$). H3 NRIs calculated with 100 bp, 1 Kb, 10 Kb and 100 Kb
bin resolution were shown for gene body regions. "r" indicates the Pearson cor-
relation coefficient in a and c. Source data are provided as a Source Data file.

replication. We found that the establishment of NRDs are not affected
by transcription inhibition (Supplementary Fig. S4f). However, when
analyzing NRS, we found that, under nascent condition, the NRS of
nascent nucleosomes became more smaller than parental nucleo-
somes after TPL treatment (compare "Pulse" conditions in Fig. 4b, c).
Under chase condition, the NRS of nascent nucleosomes became less
larger than parental nucleosomes after TPL treatment (compare
"Chase" conditions in Fig. 4b, c). Viewing the difference of NRS
between nascent and parental nucleosomes in each NRD as position
relative to the diagonal in dot plot (Fig. 4d), or as boxplot (Supple-
mentary Fig. S4g) confirmed these observations. Together, these
results suggest that TPL inhibited the wrapping of nascent nucleo-
somes. Thus, in other words, transcription promotes nascent nucleo-
some wrapping after DNA replication.

**Nucleosome wrapping domains delineate Hi-C compartments
and replication timing domains**
It has been reported that the Hi-C compartments are highly
resemble of replication timing domains[22,23]. Surprisingly, we found

that the genome-wide track of NRI is almost identical to the PC1
value[24] and the replication timing value of mouse ES cells (Fig. 5a,
Supplementary Fig. S5), with TiNRDs corresponding to A compart-
ments and early RT domains, LoNRDs corresponding to B com-
partments and late RT domains. Genome-wide correlation between
NRI, PC1 and RT value also supports the observation (Supplemen-
tary Fig. S6a), albeit with slightly higher correlation between H3 NRI
and PC1 value than correlation between H3 NRI and replication
timing value.

We used NRDs with 100 kb resolution to calculated the percen-
tage of region overlapping between NRDs, Hi-C compartments and
replication timing domains. We found that 86% of the length of A
compartments and about 80% of the length of early replication
domains overlay with TiNRDs; about 80% of the length of B compart-
ment and over 70% of the length of late replication domains overlay
with LoNRDs (Fig. 5b). Moreover, at the LoNRD-TiNRD boarders, the
Hi-C PC1 value and replication timing value sharply transit from
negative to positive; reversely, at the TiNRD-LoNRD boarders, the Hi-C
PC1 value and replication timing value sharply transit from positive to

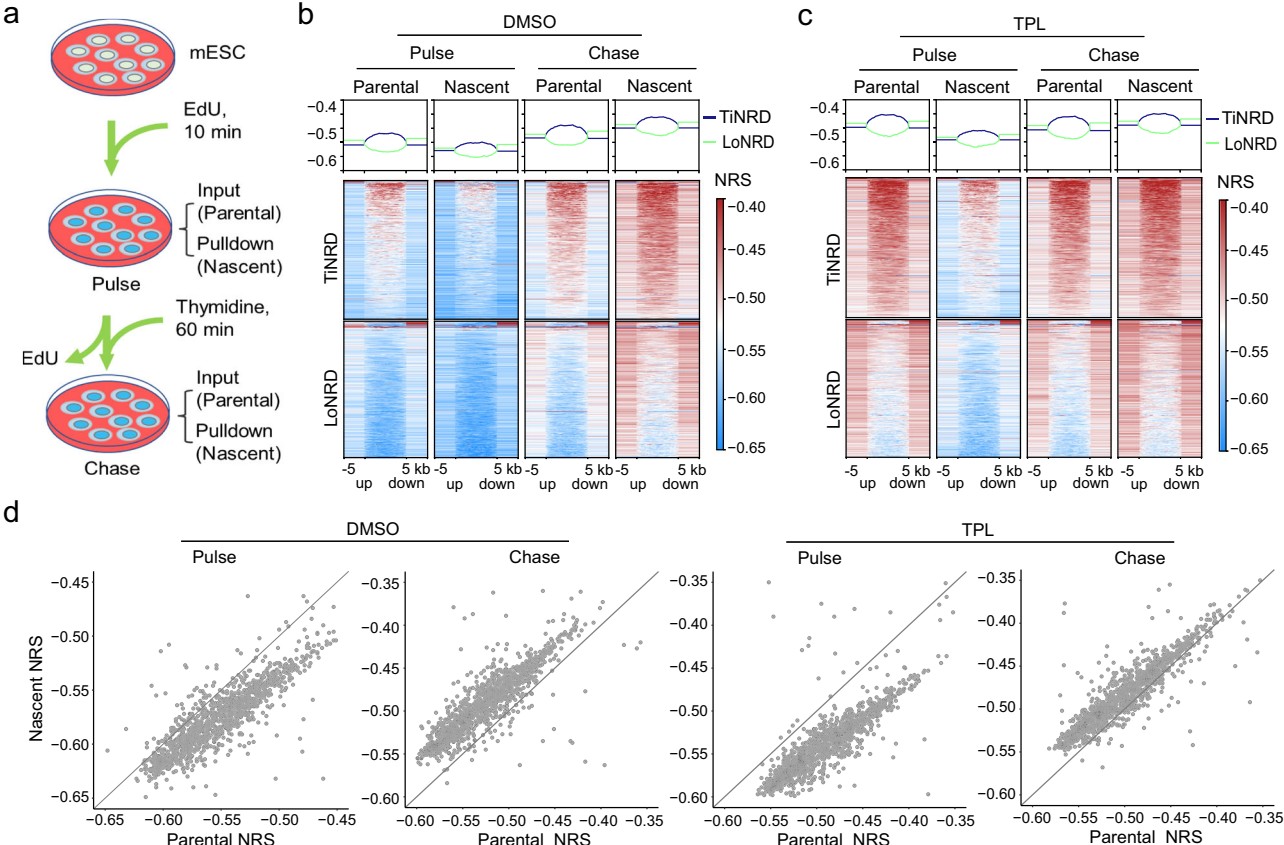

**Fig. 4 | Transcription promotes nascent nucleosome wrapping. a** Diagram show the preparation of "Pulse" and "Chase" samples for nucleosome wrapping analysis during DNA replication. "Input" contains mainly "Parental" nucleosomes, and "Pulldown" contains "Nascent" nucleosomes. Heatmaps shows MNase NRS(140)s of parental and nascent nucleosomes from "Pulse" and "Chase" conditions, without (**b**) or with (**c**) triptolide (TPL) treatment, within LoNRDs ($n = 723$) and TiNRDs ($n = 725$). TPL was used to inhibit transcription initiation, and DMSO treatment was control. MNase NRS(140) was calculated using xMNase-wrapping-seq dataset with 100 kb bins and break point at 140 bp. **d** Dot plots show the difference of NRS between parental and nascent nucleosomes as distance to the diagonal. Source data are provided as a Source Data file.

negative (Fig. 5c). Similar trend of NRI can be observed at the compartment boarders (Supplementary Fig. S6b) and replication timing boarders (Supplementary Fig. S6c). These results support the corresponding relationship between NRDs, Hi-C compartments and replication timing domains.

## Discussion

In this study, we have demonstrated that NRI can be utilized as a robust quantitative metric for characterizing the average wrapping states of nucleosomes within local chromatin regions. This quantification can be directly derived from MNase digestion sequencing data, independently of histone ChIP, and is not strictly dependent on the extent of MNase digestion. We further revealed the presence of non-uniform nucleosome wrapping states throughout the entire mouse genome, with a mutually spaced distribution of TiNRDs and LoNRDs. TiNRDs correspond to Hi-C A compartments and early replication timing domains, whereas LoNRDs correspond to B compartments and late replication timing domains. Thus, it's interesting to deduce that, as depicted in the model (Fig. 6), in active chromatin regions represented by A compartments, nucleosome wrapping is relatively tight; whereas in repressive chromatin regions represented by B compartments, nucleosome wrapping is relatively loose. However, due to the limitation of wrapping-seq, only one dimension of the physical wrapping states of nucleosomes in vivo can be deduced. Moreover, other factors beyond the strength of nucleosome wrapping may contribute to the NRS or NRI values. Other technics, such as single-molecule and

structural techniques, are needed to depict the biochemical basis of nucleosome wrapping states in vivo.

Our data suggested that nucleosome wrapping becomes more relaxed after DNA replication and gradually tightens over time. Importantly, active gene transcription promotes the wrapping of nucleosomes (Fig. 6). After transcription, nucleosomes need to be reassembled on the transcribed DNA to maintain chromatin structure. This reassembly is regulated by various histone chaperones and chromatin assembly factors[25]. One of these histone chaperones that plays a central role in nucleosome recycling during transcription is FACT (Facilitates Chromatin Transcription). FACT can bind histones and regulate the assembly or dis-assembly of nucleosomes to facilitate the passing of RNA Pol II on chromatin template[26,27]. It's recently reported that while the SPT16 subunit of the FACT complex can displace H2A-H2B dimers from the nucleosome and facilitate the formation of an open nucleosome structure, while the SSRP1 subunit holds the H3/H4 tetramer on DNA and promotes the deposition of the H2A/H2B dimer onto the nucleosome, thus maintaining the nucleosome integrity[9,10,28]. These studies indicate that FACT may regulate the wrapping dynamics of nucleosome during active transcription.

While we have initially proved that transcription activity enhances the wrapping of nucleosomes in the genic (euchromatin) regions, the mechanisms regulating the nucleosome wrapping states in heterochromatin remains to be explored. The NuRD (Nucleosome Remodeling and Deacetylase) complex is required for specific gene silencing in heterochromatin regions. Through the ATP-dependent remodeling

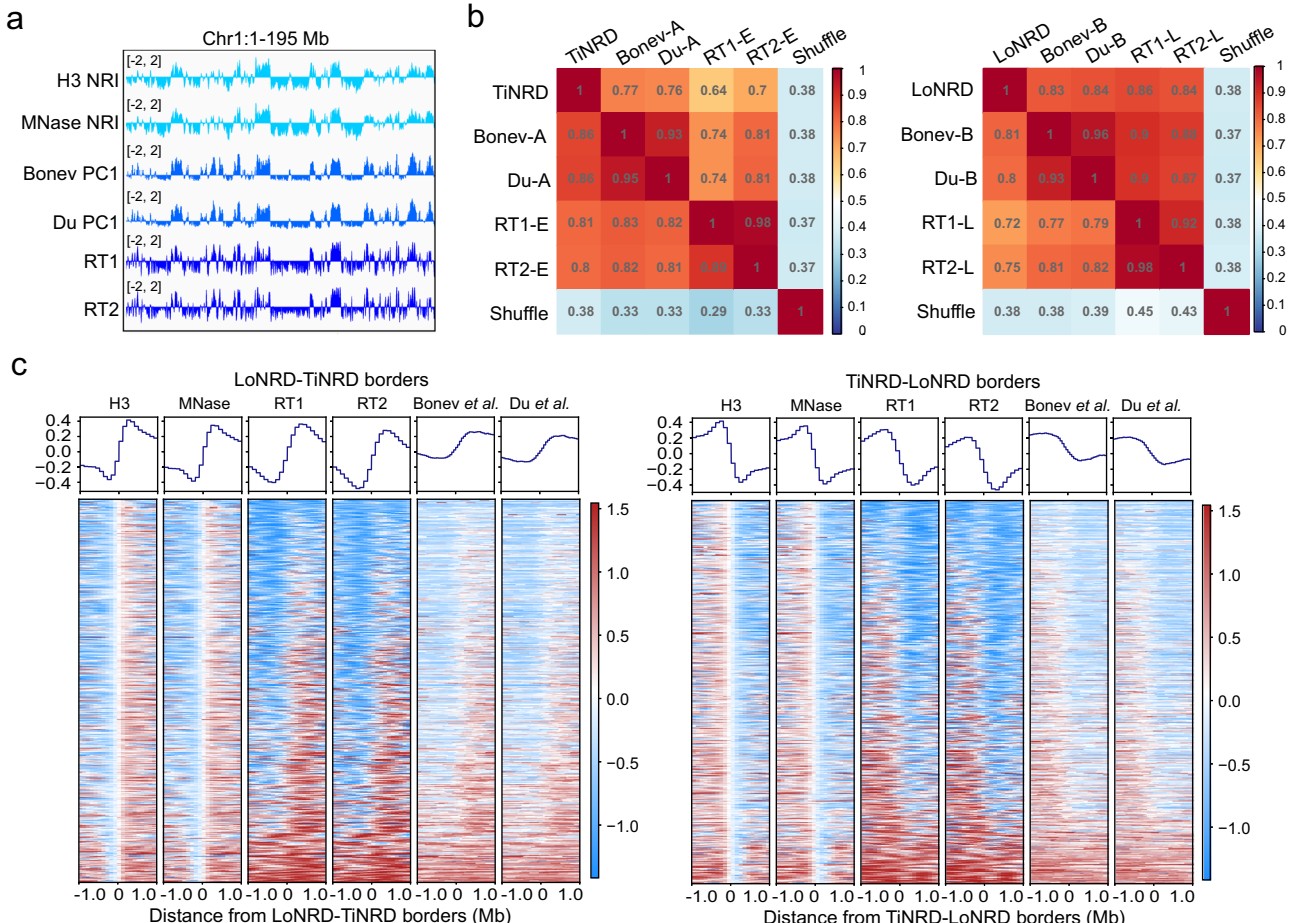

**Fig. 5 | Nucleosome wrapping domains delineate Hi-C compartment domains and DNA replication timing domains. a** IGV tracks show the pattern of H3 NRI(140), Hi-C PC1 value and replication timing values on Chromosome 1. **b** Heatmaps show the percentage of overlapping between NRDs and Hi-C compartment domains, RT domains. Values show the ratio of domain length in the column direction to domain length in the row direction. **c** Heatmaps show the distribution of H3 NRI(140), MNase NRI(140), Hi-C PC1 values and replication timing values on the 2 Mb regions around LoNRD-TiNRD borders (*n* = 725) (left panel) or TiNRD-LoNRD borders (*n* = 725) (right panel). Source data are provided as a Source Data file.

activity of the CHD subunit, it alters chromatin structure and DNA accessibility to regulate gene transcription[29]. CHD3 is one of the nine members of CHD family of ATPases (CHD1–9). Binding of CHD3's tandem PHD fingers to histone H3 tails, which can be enhanced by H3K9me3 modification, promotes the unwrapping of nucleosomes in vitro[30]. Heterochromatin protein 1 (HP1) is implicated in the formation and maintenance of heterochromatin via interaction with H3K9me3 modified nucleosomes[31]. Binding of multiple Swi6 (the *S. pombe* HP1 protein) molecules to nucleosomes can promote HP1-chromatin LLPS (liquid–liquid phase separation), through increasing the solvent exposure of buried nucleosomal core histones[32]. This HP1 mediated nucleosome reshaping process intrinsically resembles nucleosome unwrapping, indicating that HP1 binding may promote DNA unwrapping from the nucleosome core. While heterochromatin is characterized by its repressive nature, it is important to recognize that it is not a static and entirely silenced region of the genome. Instead, it is dynamically transited between repressive and active states, which involves various regulatory elements and factors. Thus, we speculate that the relative loose wrapping state of nucleosomes in the heterochromatin regions will allow controlled accessibility of regulatory factors to their target binding sites, during various cellular activities, such as gene expression regulation during development.

In summary, our research uncovered a previously unidentified domainization principle of the chromatin, which originates from the unit of the primary structure of chromatin—the nucleosome itself, but intriguingly gives rise to nucleosome wrapping domains that precisely coincide Hi-C compartment domains and DNA replication timing domains. This concept invokes that the protein tertiary structure is encoded in the amino acid sequences.

## Methods

### Cell culture

Mouse ES cell (R1) was a kind gift from Dr. Guohong Li, Institute of Biophysics, CAS. R1 cells were cultured in the medium with 80% DMEM (Thermo, 11965092), 15% FBS (Hyclone, SH30070.03), Nonessential amino acids (EmbryoMax, TMS-001-C), 2-Mercaptoethanol (EmbryoMax, ES-007-E), L-glutamine (EmbryoMax, TMS-002-C), Nucleosides (EmbryoMax, ES-008-D), Pen/Strep (EmbryoMax, TMS-AB-2C) and 1000U/ml leukemia inhibitory factor (LIF) (ESGRO, ESG1107) in standard incubator with 5% $CO_2$ at 37 °C. For wrapping-seq, mouse ES cells plated for about 48 h were directly crosslinked with 1% formaldehyde in PBS for 10 min at room temperature, then quenched by 125 mM glycine, and scraped off from the dish with a cell scraper. Then the cells were washed with cold PBS for twice and the cell pellet were stored at −80 °C.

To label nascent nucleosome, EdU (5-ethynyl-2′-deoxyuridine) was added at 20 μM final concentration and cultured for 10 min. Then the medium with EdU was drained off, and the cells were washed with 1XPBS pre-warmed at 37 °C for twice: to prepare nascent sample, the cells were crosslinked directly as described above for wrapping-seq; to

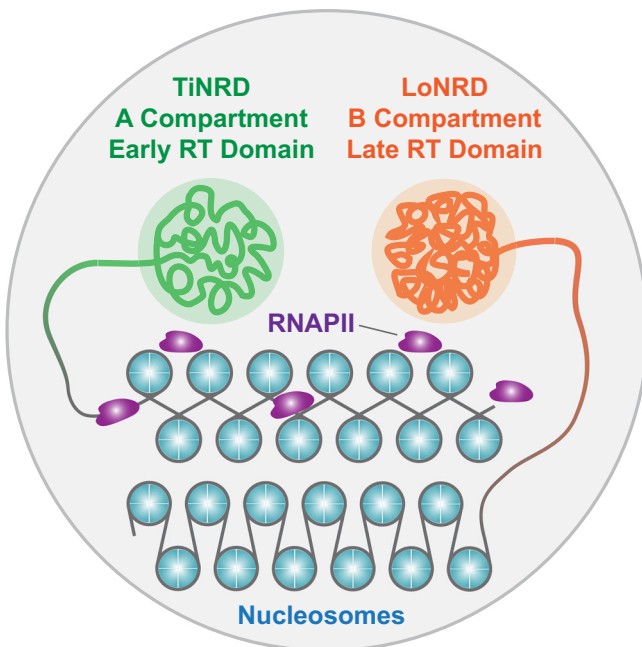

**Fig. 6 | Nucleosome wrapping states encode principles of 3D genome organization.** Working model shows that genome of mouse ES cells is organized into nucleosome wrapping domains (NRDs), including TiNRDs and LoNRDs. TiNRDs and LoNRDs represents genomic regions enriched with relatively tightly or loosely wrapped nucleosomes, respectively. TiNRDs overlap precisely with Hi-C A compartment domains and early replication timing (RT) domains. LoNRDs overlap precisely with Hi-C B compartment domains and late RT domains. Thus, our model reveals a previously unrecognized connection between genome organization at nucleosome array level and higher order level. Moreover, our model suggests that nucleosomes in active chromatin environment (A compartment) wrap tighter than nucleosomes in inactive chromatin environment (B compartment), as a consequence of transcription promoted nucleosome wrapping.

prepare chase samples, the cells were further culture with mouse ES medium with 10 µM thymidine for 60 min, then crosslinked directly as described above for wrapping-seq. To inhibit transcription initiation, mouse ES cells were cultured with 1 mM triptolide (Sigma, T3652) final concentration or equal volume of DMSO only as control for 45 min before EdU labeling, then triptolide or DMSO was added back during chasing.

### Wrapping-seq

**Wet lab experiments.** Cell pellet of ~1 × 10^7 crosslinked cells was resuspended in 1 mL nuclei extraction buffer (10 mM HEPES, PH 7.5; 300 mM NaCl; 3 mM MgCl$_2$; 0.5% IGEPAL CA-630; 0.1% SDS) with EDTA-free protease inhibitor cocktail (Roch, 05892791001) and incubate on ice for 10 min. Then cell pellet was washed once with buffer A (10 mM Tris, PH 7.5; 10 NaCl; 60 mM KCl; 3 mM MgCl$_2$) with EDTA-free protease inhibitor cocktail. For MNase digestion, cell pellet was then resuspended in buffer A with 2 mM CaCl$_2$ and 0.4 U/mL MNase (Sigma, N3755), and incubated at 37 °C on a metal incubator shaking at 900 rpm for 30 min or as indicated for time-course digestion. Final concentration of 10 mM EDTA and 1% SDS was added to stop the digestion.

For MNase-seq, the digested chromatin was crosslink reversed by heating at 65 °C for 6 hours, and DNA were extracted using a standard phenol-chloroform extraction procedure. For paired-end sequencing, libraries without size selection were prepared as described in ref. 15 using NEBNext Ultra DNA Library Prep Kit for Illumina (E7370L) and were sequenced on a DNBSEQ-T7 (MGI) platform with PE150 model.

For H3 or H4 ChIP, the digested chromatin was sonicated using a Q800R3 Sonicator (Qsonica), and the chromatin particles before crosslink reversion are about 1000–2000 bp after resolved on native 1% agarose gel, which is the typical chromatin particle size for ChIP-seq. Then, chromatin was first incubated with 2 µg H3 (Abcam, ab1791) or H4 (Cell Signaling Technology, 14149S) antibody in RIPE-150 (50 mM Tris-HCl, 150 mM NaCl, 1 mM EDTA, 0.5% Triton X-100, protease inhibitors) at 4 °C, then the BSA blocked protein A/G Dynabeads were added and incubated overnight at 4 °C. The Dynabeads were washed by RIPE-150 with 0.1% SDS for 5 times, and eluted with Direct Elution Buffer (10 mM Tris-HCl pH8, 0.3 M NaCl, 5 mM EDTA pH8, 0.5% SDS). The chromatin was crosslink reversed by heating at 65 °C for 6 h, and DNA were extracted using a standard phenol-chloroform extraction procedure. For paired-end sequencing, libraries without size selection were prepared as described in ref. 16 using NEBNext Ultra DNA Library Prep Kit for Illumina (E7370L) and were sequenced on a DNBSEQ-T7 (MGI) platform with PE150 model.

### Sequence data alignment

Paired-end reads were trimmed for adaptor sequence using cutadapt v4.3[33] with parameters: -a AGATCGGAAGAGCACACGTCTGAACTCC AGTCAC -A AGATCGGAAGAGCGTCGTGTAGGGAAAGAGTGT -e 0.1 -n 2 -m 35 -q 30 –pairfilter = any, and then mapped to mm10 using Bowtie2 v2.5.1[34] with parameters: -I 10 -X 1000 −3 5–local–no–mixed–no–discordant–no–unal. Duplicates were marked using picard MarkDuplicates v2.27.5 (https://broadinstitute.github.io/picard/) with default parameters and removed using samtools view v1.17[35] with parameters: -f 2 -F 1024 -q 10. Unique pair-end reads in bam format was converted to bed format, and then grouped according to fragment length using a custom python script.

### Nucleosome wrapping score (NRS) or Nucleosome wrapping index (NRI) calculation

Mouse genome was segmented in to continuous non-overlapping 100 kb (or 100 bp, 1 kb, 5 kb, 10 kb, 50 kb) bins. Then, within each genomic bin, fragments from each fragment length group were counted by annotateBed from bedtools v2.30.0[36]. Fragment length groups used includes 50–80 bp, 80–90 bp, 90–100 bp, 100–110 bp, 110–120 bp, 120–130 bp, 130–140 bp, 140–150 bp, 150–160 bp, 160–250 bp or 50–140 bp, 140–250 bp. Then when a fragment length break point ($X$ bp) is chosen, nucleosome wrapping score (NRS) is computed as the relative deviation between the number (a) of fragments within $X$ ~ 250 bp and the number (b) of fragments within 50 ~ $X$ bp, expressed as NRS($X$) = (a − b)/(a + b). Then the genome-wide NRS was transformed as z-score to derive genome-wide nucleosome wrapping index (NRI).

$$Z(i) = \frac{X(i) - \mu}{\sigma} \tag{1}$$

In Eq. (1), $Z(i)$ is the z-score of the NRS of the i-th genomic interval, $X(i)$ is the NRS of the i-th genomic interval, μ is the mean of NRS of all genomic intervals, and σ is the standard deviation of NRS of all genomic intervals.

For IGV (Integrative Genomics Viewer)[37] visualization, NRS or NRI was Loess-smoothed by 10 bins using R function loess() from R package 'stats'. Un-smoothed NRSs and NRIs were used for annotation of genomic features.

### Nucleosome wrapping domain (NRD) detection

The command FindHiCCompartments from homer2[38] was used to detect tight NRD (TiNRD) based on the 10 bins smoothed 100 kb bin NRI (140 bp) data of H3 with default parameter; and the parameter "-opp" was used to output the loose NRD (LoNRD). TiNRDs and

LoNRDs were genomic domains with continuous positive or negative NRI, respectively. Genomic features with NRDs were annotated using annotatePeaks.pl from homer2[38]. Heatmaps were produced using plotHeatmap from deeptools v3.5.1[39] for NRI or NRS visualization in NRDs or genic regions.

### Replication timing profiling for mouse ES cells

Replication timing profiling and timing domain for mouse ES cells was performed as described[40,41]. Briefly, mouse ES cells were pulse labeled with 100 µM BrdU (5-Bromo-2′-deoxyuridine) for 2 h, then trypsinized to single cell suspension and fixed with ice-cold ethanol. Then the cells were stained with Propidium iodide and cells in S-phase were FACS sorted into four fractions (S1, S2, S3, S4) according to Propidium iodide signal. Then genomic DNA was prepared for each sample via a standard phenol-chloroform extraction procedure, and then sonicated via a Q800R3 Sonicator (Qsonica) to average 200 bp fragment size. NGS adaptor were ligated using NEBNext Ultra DNA Library Prep Kit for Illumina (E7370L). Then the DNA was denatured by heating at 95 °C for 5 min and immediately cooled on ice for 2 min. Nascent DNA strands were enriched via pulling down by anti-BrdU antibody (BD Pharmingen, cat. no. 555627). Library were amplified and sequenced on a DNBSEQ-T7 (MGI) platform with PE150 model. Sequence data alignment was performed as described above.

Sequencing data were aligned to mouse genome as described above. To calculate genome-wide replication timing, mouse genome was binned into non-overlapping 50 kb genome windows. Within each window, fragments from each sample were counted by annotateBed from bedtools v3.5.1[39]. After normalizing the total counts from each sample to 1 million, replication timing were calculated as log2((S1 + S2)/(S3 + S4)), and quantile normalized with R function normalize.quantiles.use.target() from R package 'preprocessCore'. The quantile normalized replication timing value were used for replication timing domain detection via FindHiCCompartments from HOMER[38]. Heatmap was produced using plotHeatmap from deeptools v3.5.1[39]. For IGV (Integrative Genomics Viewer)[37] visualization, the normalized value was Loess-smoothed using R function loess() from R package 'stats'.

### Reporting summary

Further information on research design is available in the Nature Portfolio Reporting Summary linked to this article.

## Data availability

High throughput sequencing data generated in this study have been deposited to Gene Expression Omnibus (GEO) database under accession code GSE243091 (https://www.ncbi.nlm.nih.gov/geo/). Source data are provided with this paper. Drosophila MINCE-seq fastq data was downloaded according to GEO record GSE76120, and mapped to drosophila genome version dmel-r6.16 (https://flybase.org). Samples GSM1974516 and GSM1974518 were merged as nascent input, and used for NRI visualization and NRD detection with 100 kb bin resolution as shown in Fig. 2d. Samples GSM1974517 and GSM1974519 were merged as nascent pulldown. Samples GSM1974528 and GSM1974530 were merged as chase input. Samples GSM1974529 and GSM1974531 were merged as chase pulldown. NRS was calculated for nascent input, nascent pulldown, chase input and chase pulldown samples as shown in Supplementary Fig. S4e. S. cerevisiae MNase-seq fastq data was downloaded according to GEO record GSE30551, and mapped to yeast genome GCF_000146045.2 (https://www.ncbi.nlm.nih.gov). Sample SRR3193265 were used for NRI visualization and NRD detection with 1 kb bin resolution as shown in Fig. 2c. H3K4me1, H3K4me3, H3K9me3, H3K27me3 and Polr2a ChIP-seq bam files and peak files, ChIP input bam files and total RNA-seq files of mouse ES cell line Bruce4 were downloaded under accession ENCSR343RKY from https://www.encodeproject.org. DNase-seq bigwig signal file and peak file of

mouse ES cell line E14 were downloaded under accession ENCSR000CMW from https://www.encodeproject.org. Hi-C compartments coordinates and PC1 values of mouse ES cell was downloaded under accession 4DNESUCLJAZ8[42] and 4DNESMXBLGKA[24] from https://data.4dnucleome.org. Source data are provided with this paper.

## Code availability

The Jupyter notebook used to calculate nucleosome wrapping score and nucleosome wrapping index can be found at https://github.com/WenZengqi/nucleosome-wrapping-index and archived in Zenodo[43].

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

## Acknowledgements

This work was supported by the Ministry of Science and Technology of China (2022YFA1302801), the National Natural Science Foundation of China (32370647) and Shenzhen Science and Technology Program grant (RCYX20221008092930079) to HL. This work was also supported by Shenzhen Science and Technology Program grant (GXWD20201231165807008, 20220817134430001) to ZW.

## Author contributions

Z.W. and H.L. designed and supervised the research; Z.W. performed experiments and bioinformatic analysis of wrapping-seq; R.F. and F.Z. performed replication timing experiments under the supervision of H.L.; R.Z. and X.Y. performed BrdU chasing experiments under the supervision of Z.W.; Z.W. and H.L. prepared the manuscript with contributions from all authors.

## Competing interests

The authors declare no competing interests.
