## [Transparent Peer Review file · Nature Communications]

Nucleosome wrapping states encode principles of 3D genome organization

Corresponding Author: Dr Zengqi Wen

Version 0:

Reviewer comments:

Reviewer #1

(Remarks to the Author)

The authors used the MNase-X-ChIP-seq protocol to map the lengths of DNA fragments resulting from MNase enzyme cleavage across genome-wide. Utilizing this data, they introduced metrics, namely the Nucleosome Wrapping Score (NRS) and Nucleosome Wrapping Index (NRI), to describe the average wrapping states of nucleosomes within specific genomic regions. Upon analyzing the distribution of nucleosome wrapping states across genome-wide, the authors identified a very intriguing phenomenon: the genome can be partitioned into tightly wrapped nucleosome wrapping domains (TiNRDs) and loosely wrapped nucleosome wrapping domains (LoNRDs); TiNRDs and LoNRDs precisely coincide with A compartment domains and B compartment domains, respectively. I have the following concerns:

Major

1. The research community dedicated to investigating 3D chromatin organization broadly agrees that chromosomes are hierarchically organized into large A/B compartments, further divided into smaller topologically associating domains (TADs). The work here presented, claims to reveal a novel level of chromatin organization. However, it simultaneously emphasizes that TiNRDs and LoNRDs precisely coincide with widely studied A and B compartments, respectively. Thus, the clarity of the novel level of chromatin organization is needed.
2. The work here presented, also claims to reveal a novel principle of chromatin organization. However, the investigation primarily focuses on the wrapping length of DNA on nucleosomes, uncovering a novel phenomenon or feature within euchromatin and heterochromatin – specifically, that the wrapping lengths of DNA on nucleosomes within euchromatin are likely longer than those within heterochromatin. Despite this revelation, the study does not establish a causal relationship between this feature and the formation of NRDs or compartments. Further clarification is necessary to elucidate the proposed novel principle of chromatin organization.
3. Fig. 3a displays a clear positive correlation between H3 NRI(140) and DNase I sensitivity, while Fig. 1d reveals a similar correlation between H3 NRI(140) and H3 ChIP signals. This indicates that chromatin regions with high DNase I sensitivity (indicating low nucleosome density) also exhibit high H3 ChIP signals (indicating high nucleosome density). To enhance the study's credibility, the author should provide an explanation for this apparent contradiction.
4. In the section titled 'Transcription promotes nascent nucleosome wrapping,' the author labels nascent nucleosomes using EdU and subsequently conducts a wrapping-seq experiment to explore nucleosome wrapping dynamics during DNA replication. In this section, it is crucial for the author to initially rule out the influence of EdU labeling on the distribution of fragment length (as shown in Fig. 1a) and the calculation of NRI/NRS values. Without such exclusion, it becomes challenging to determine whether the observed differences in NRS values are attributed to DNA replication or EdU labeling.
5. In Fig. 1e, certain chromatin regions exhibit a notable change in NRS under different digestion time, while NRI remains relatively constant. Consequently, I recommend that the author uses NRI instead of NRS as a metric for assessing

nucleosome wrapping states in the section 'Transcription promotes nascent nucleosome wrapping' and in Fig. 4. Considering the author's claim that 'Input' (parental) and 'Pulldown' (nascent) nucleosomes in each 'Nascent' or 'Chase' condition undergo the same MNase digestion, it is expected that NRI values would yield similar results.

6. In Fig. 4d, the author employs a bin size of 100kb for the NRS calculation, which is close to the size of DNA replication domains. It's worth noting that chromatin is typically partitioned into TADs with a median size of 185kb (as reported in Cell 159, 1665-1680), and TADs are recognized as stable units of replication timing regulation (as reported in Nature 515, 402-405, 2014). Given this context, I recommend that the bin size used to measure nucleosome wrapping states during the DNA replication process be much smaller than the size of replication domains.

Minor

7. As shown in Figure S1e, the NRI values vary with respect to the size of each bin when computing the NRS values. It is necessary to specify the bin sizes of NRI values used in other figures in this work.

8. Does the "MNase input" in Fig.1d mean classical MNase-seq data or MNase-seq data from wrapping-seq? Please use some terms to distinguish them.

9. One type error in line 175 "Correspondingly, quantification of NRS in these peak regions showed that nucleosomes in H3K4me1 or H3K4me3 peaks regions have higher H3 NRS(140) that(should be than) those in H3K9me3 or H3K27me3 peaks regions (Fig. S3a)."

Reviewer #2

(Remarks to the Author)

In this paper, the authors utilize a method they had previously developed termed MNase-X-ChIP-seq to characterize nucleosome wrapping genome wide. This pipeline, termed wrapping-seq, allows the authors to generate a nucleosome wrapping score (NRS) using the relative balance of fragment sizes at discrete genomic bins. This wrapping score is then converted to a z-score, which the authors term the nucleosome wrapping index (NRI). Using the NRI, the authors provide data suggesting that nucleosome wrapping varies significantly across the genome, with euchromatin regions showing much higher wrapping when compared to heterochromatic regions. The authors further demonstrate that newly replicated regions show lower levels of wrapping when compared to steady-chromatin, consistent with previous observations that nascent chromatin is hypersensitive to MNase digestion. Interestingly, the authors show that transcription plays an active role in promoting DNA wrapping, as inhibition of RNA pol II after replication fork passage prevents nascent chromatin from acquiring steady-state levels of nucleosome wrapping. Lastly, authors show that tightly-wrapped domains (TiNRD) and loosely-wrapped domains (LoNRDs) correlate with A and B compartments identified by Hi-C. These observations are interesting and reveal potentially important insights into differences in chromatin structure across distinct domains. However, there are several key issues that must be addressed that are described hereafter:

Major concerns:

1. No mention of MNase sequence bias. MNase has a known sequence bias, in that it digests AT-rich DNA preferentially over GC-rich DNA¹. Sequence composition varies significantly across the genome in a manner that could impact the NRI measurements calculated by the authors. Authors should discuss this bias and demonstrate that sequence biases do not strongly contribute to the observations made in this manuscript, particularly the strong NRI differences noted between euchromatin and heterochromatin.
2. The authors claim that because NRI patterns from different digestion time points are highly similar, MNase-seq NRI(140) is robust for nucleosome wrapping detection, despite of a certain degree of digestion variation. However, their time-course is not sufficient to be able to make these claims. Previous studies have shown significant variation in nucleosome recovery from 1-10 minutes of MNase treatment². Indeed, many of the most unwrapped, fragile nucleosomes, could be depleted from libraries after 10 minutes of treatment^{2,3}. Authors should evaluate NRI140 after 1 and 5 minutes MNase digestion to fully validate claims that: "NRI140 is robust for nucleosome wrapping detection."
3. The unusual correlation of NRI values to ChIP yield is strange, as it suggests that some aspect of the authors' workflow is losing signal in heterochromatic areas. The authors do a nice job addressing a potential cell cycle-related explanation, but the issue remains unresolved. One concern is that the heterochromatic regions are resistant to both MNase and sonication³. Authors should that their MNase treatment conditions are able to sufficiently fragment and solubilize genomic DNA to allow for a fair comparison between euchromatin and heterochromatin.
4. The manuscript should be carefully proofread and edited for clarity and proper grammar.

Minor points:

- (1) In figure S1K the G2-phase data is inappropriately labeled "S-phase"

References:

- 1 Dingwall, C., Lomonosoff, G. P. & Laskey, R. A. High sequence specificity of micrococcal nuclease. *Nucleic Acids Research* 9, 2659-2674, doi:10.1093/nar/9.12.2659 (1981).
- 2 Chereji, R. V., Bryson, T. D. & Henikoff, S. Quantitative MNase-seq accurately maps nucleosome occupancy levels. *Genome Biology* 20, 198, doi:10.1186/s13059-019-1815-z (2019).
- 3 Mieczkowski, J. et al. MNase titration reveals differences between nucleosome occupancy and chromatin accessibility.

Reviewer #3

(Remarks to the Author)

The authors use MNase-digested DNA fragment lengths as a genome-wide measure of DNA unwrapping: Smaller fragments mean more unwrapping. This is seen across large swaths of eukaryotic genomes. More wrapping (longer DNA fragments) is seen in gene bodies versus elsewhere. Certain histone modifications and Pol II also track with gene bodies, and so the same results are seen where they are enriched. Unfortunately, essentially all the key results can be explained by the well-known A/T-sequence bias of MNase and the nonrandom distribution of A/T and G/C across eukaryotic genomes. The authors completely ignore this. Even the author's prior work in this area (Ref. 15) ignores this reality. This must be addressed definitively before this work can be meaningfully considered for publication.

Specific comments

L. 101-122: NRS is a metric of wrapping (and NRI is a statistical derivative, Z-score). However, this initial explanation in the beginning of the results is largely opaque to the reader. I had to go through the calculation simply to understand what was being done, but then came away with a metric that, at least to me, had uncertain meaning. The reader would be better served by having some text-based meaning of what, at the end of all this, is actually being measured. It appears to be there at the end of the first paragraph, and so it would be more helpful to move this up to the beginning, as a rationale as to why one chooses such a metric, and its caveats.

L123: "Bin lengths" is unclear. I think "genomic intervals (e.g., 1 kb, 10 kb, etc.)" would be clearer.

Fig 1f. Shouldn't the axis be flipped (and same with all other equivalent plots), with ChIP being the dependent variable? Also, I was not convinced that the positive correlation of NRI with H3 ChIP signal was due to differences in H3 density, as opposed to technical differences (e.g., in chromatin extraction, ChIP efficiency, library construction, PCR, and depth of sequencing). This has relevancy implications for Fig. 1g/S1K, making it moot. Fig. 1g is not interpretable by a reader. Fig. S1K has a typo (S phase). The conclusions from both are not well grounded. I'm not sure that the proposed problem that Fig. 1g was meant to address is even valid. Even in S phase the vast majority of the genome has a normal density of nucleosomes. So I don't see how this is a concern.

L156. At this point the author draw the conclusion of what NRS/NRI is measuring is in fact robust. It may be robust, but I am not convinced that the A/T-rich bias of MNase isn't driving the patterning. The authors should also plot A+T density, along with the NRI, and provide convincing data and analyses that rule out sequence bias of MNase digestion.

L175: mES have high GC-content in gene bodies making them more MNase resistant (high NRS) relative to intergenic regions. Yeast have high A/T-content in intergenic regions making the gene bodies relatively more MNase resistant. Thus, the correspondence between yeast and mouse is not surprising, and completely explainable by DNA sequence.

L182+: Since histone modifications and Pol II occupancy follow gene bodies, then MNase sequence bias explains these results as well.

L217: Subtitle is inappropriate for the paragraph that follows.

L422: There is no description of how the NRI was calculated. All I could find was "Then the genome-wide NRS was transformed as z-score to derive genome-wide nucleosome wrapping index (NRI)." Therefore no independent assessment of what NRI is measuring could be determined.

Version 1:

Reviewer comments:

Reviewer #1

(Remarks to the Author)

I would like to thank the authors for their response to my concerns. However, I still have one final query regarding the interpretation of the output of the MNase-X-ChIP-seq experiment.

In the manuscript, the authors state that they use the MNase enzyme to cleave DNA and utilise the lengths of DNA fragments to calculate NRS/NRI. The larger NRI values observed in H3-wrapping-seq suggest that DNA wraps more tightly on nucleosomes. This claim is based on the supposition that DNA is stably wrapped on nucleosomes and that DNA fragments are protected by histones from MNase cleavage. However, as stated in lines 45-47, DNA located on nucleosomes can rapidly transition between an "unwrapping" and "rewrapping" state. It is therefore reasonable to speculate that in the MNase enzyme digestion process, DNA fragments may also transition from histone wrapping state to histone unwrapping state. If the aforementioned speculation is indeed accurate, it can be argued that DNA regions with a higher NRI value indicate that histone proteins exhibit a preference for long DNA fragments from this region. It would be beneficial to ascertain whether there is any evidence to disprove this later speculation.

(Remarks on code availability)

Reviewer #2

(Remarks to the Author)

In this revised version of the manuscript, the authors have satisfactorily addressed the major concerns. The authors should further proofread the manuscript to address remaining grammatical errors.

(Remarks on code availability)

Reviewer #3

(Remarks to the Author)

I appreciate the reviewers attempt to address the reviewer's concerns and revising the manuscript. The manuscript reads more clearly, and allows us to examine it more thoroughly. The biggest sticking point was that the results could be explained by the A/T-bias of MNase. The new analysis does not change this, and in fact seems to provide evidence that the results can indeed be explained by MNase sequence bias.

Main comments on A/T bias

1. The authors find that there is high A/T content at cleavage sites (Fig. S2f), which only reconfirms the intrinsic sequence bias known for MNase. As the author point out, the A/T bias at the cleavage site is the same for Lo vs Ti NRDs. This is expected because at the cleavage site, one is simply measuring the A/T bias that is intrinsic to cleavage by MNase. It is not measuring the propensity to form Lo vs Ti NRDs. This can be found in the flanking genomic sequences that are not protected. Indeed, they find high A/T content in the flanking regions of LoNRDs compared to TiNRD, which is what one would expect if the LoNRD vs TiNRD pattern is driven by A/T bias. Since the flanking region outside of the protected region is more sensitive to MNase, one expects more cleavage opportunities and thus a higher proportion of LoNRDs over TiNRDs. One can discuss this back and forth, but the real answer resides in a control experiment, which has not been done. For example, if naked DNA is digested with MNase to the same size range, what would the NRI map look like when analyzed side-by-side with chromatin

2. The analysis in Fig 2e, 2f does not seem appropriate for addressing the MNase bias because the authors are looking at the A/T sequence content of protected fragments, rather than the presumably unprotected DNA located immediately adjacent. Also, it is not clear how plotting fragment length vs A/T frequency addresses the question. Protected fragments regardless of length should have the same A/T content, otherwise MNase would preferentially digest the A/T-rich ones, thereby depleting A/T from the population.

3. Fig. 4g indicates that higher NRI (more "fully wrapped") nucleosomes have low A/T content. The correlation is -0.35. I think it is arbitrary as to whether one sees this correlation as being weak or not weak. Nonetheless, this trend indicates that an A/T bias of MNase cannot be excluded as a likely explanation for all the results.

4. The relationship of "tighter wrapping" with transcription is consistent with transcription occurring at CpG islands in mammals which are inherently more resistant to MNase (due to A/T cleavage bias) and thus will have their "wrap" skewed to longer fragment lengths. The authors should conduct the same analyses on their *Drosophila* and *Yeast* data, which do not have CpG islands.

Other major comments

5. One would need to control for variable chromatin/DNA extraction efficiency from cells. This could be examined with the whole genome sequencing data generated for Fig. 1h. The authors note that NRI correlates with histone H3 ChIP signal. Could this mean that Lo NRD portions of the genome (B compartments?) extract less efficiently (possibly due to crosslinking)? If so, then small fragments might extract relatively more efficiently in such regions, which would leave larger fragments behind.

6. Browser plots of NRI vs genomic coordinate along the x-axis (eg. Fig. 1c and more) lack a definitive y-axis range, such that one cannot evaluate the NRI values (Z-scores). It seems to me that the Z-scores barely reach 1, which means that across large genomes, these events will occur frequently by random chance. Fig. S2a-c reveals that the average NRI for an NRD bin is less than 0.5, which if I understand its meaning, is statistically not significant. I also wonder how smoothing affects correlations (Methods: "10 bins smoothed 100 kb bin NRI (140bp) data"). For example, a correlation scatter plot will inappropriately have a better correlation if the data are smoothed. Further, it is not clear how the values are tallied in browser windows (e.g., Fig. 1c). What happens when the quantity of values exceeds the number of pixels on the graph? Are they averaged? When reporting the number of LoNRD vs TiNRD as in Fig 2b, statistics need to be reported, such as a violin plot of NRI values. This exists somewhat in Fig. S2a-c, but there is no discussion of how meaningful an average NRI (Z-score) of ~0.5 is.

7. Line 137 states that there is obvious variation of the MNase-X-ChIP signal across the genome (Fig. 1d). However, this would only be obvious in the context of a negative control, so that we can see what is a "mountain" and what is a "molehill".

Without a robust statistical tests for NRD definition (as opposed to NRS), it is difficult to know how real these peaks are.

Minor comments

8. Line 129. It is not likely that mononucleosomes are the dominant digestion product, compared to all others as a group, which is the appropriate comparison group. Mono vs non-mono needs to be quantified.

9. It is very difficult to figure out which species is being used in some experiments. Fig. 2b as one example.

10. Line 143. "we also need to figure out whether the variation of nucleosome density is due to experimental error of wrapping-seq or due to unsynchronized DNA replication in a cell population." This seems like a false dichotomy. What exact "experimental error" is the reviewer referring to? How does "unsynchronized DNA replication" have anything to do with this? The data are from cell population averages, and so any potential cell cycle issue should be averaged out.

(Remarks on code availability)

Version 2:

Reviewer comments:

Reviewer #1

(Remarks to the Author)

I appreciate the authors addressing my concerns regarding the unwrapping and rewrapping process of DNA during MNase enzyme digestion. It is now clearer to me. However, upon reviewing the comments and responses from the second reviewer, I think about the role of MNase's A/T bias on NRI value. Surprisingly, I found that an arbitrary model involving MNase A/T bias alone can explain the conserved pattern of NRI without the role of nucleosome wrapping. Details are elaborated below: To evaluate the role of A/T bias in the determining NRI values, I constructed an arbitrary model as below. Suppose I have a naked DNA chain of 10 Mb (with no nucleosomes bound), divided into 100 100-kb DNA bins. Each 100-kb bin is composed of 400 250-bp fragments. I can estimate the number of long fragments (a) and short fragments (b), as noted in Fig. 1a of the manuscript. Before MNase digestion ($t=0$), $a=400$ and $b=0$ for every 100-kb bin. When MNase cuts the DNA into pieces, the value of a decreases while b increases. Assuming MNase digestion process is randomly for each 100-kb bin, a(t) would decay exponentially with time t. Therefore, I can roughly express a(t) as $400 \cdot \exp(-\lambda t)$ and b(t) as $400 \cdot (1 - \exp(-\lambda t))$, where λ is a parameter related to cleavage efficiency, with each 100-kb bin having a specific λ value.

For each 100-kb bin, I first randomly assigned λ values within the range of 0.1 to 0.3. Then I calculated a(t) and b(t) as well as NRS and NRI for all 100-kb bins at time $t=1$. Subsequently, I recalculated the NRS and NRI for all bins at additional time points, such as $t=2, 3, 4,$ and 5 . Comparing the NRS and NRI values at different time points, I found that the NRI value for each 100-kb bin remained almost conserved, similar to the results observed in the manuscript. Considering that we only involved cleavage efficiency in the estimation, the conserved NRI values challenge the speculation that higher NRI values represent tighter DNA wrapping on nucleosomes, as well as the conclusions of the manuscript.

Based on the results from the arbitrary model illustrated above, I strongly suggest that the authors provide validated evidence to demonstrate that higher NRI values are caused by tighter nucleosome wrapping, rather than other factors such as MNase A/T bias. Otherwise, the conclusion in the manuscript will not be solid.

(Remarks on code availability)

Reviewer #2

(Remarks to the Author)

In this latest revised version of the manuscript, the authors further respond to reviewer 3's concerns that nucleosome-wrapping scores (NRI) may be the result of underlying AT-bias in MNase digestion. Reviewer 3 proposes new analysis and a potential experiment to try and assess the impact of MNase on NRI. Reviewer 3 argues that rather than plotting AT content of protected fragments (Figure 2g), the authors should plot AT content of immediately adjacent (digested) regions to better assess the correlation between AT content of digested regions and protected fragment size. However, the results of this analysis may not be useful in definitively determining the contribution of MNase bias to the authors data because of the higher incidence of AT content in heterochromatin areas with low Pol II. As the authors provide evidence that Pol II levels are strongly correlated with NRI, it would be expected that heterochromatin regions with high AT content should show low NRI, even if underlying sequence had no impact on wrapping score. Therefore, even if the analysis proposed by reviewer 3 were to show high AT content at regions of low wrapping, it would not be clear whether this is due to low transcription or MNase bias. I also agree with the authors that the experiment proposed by the reviewer 3 to digest naked DNA with MNase would not be helpful in determining the contribution of AT bias to NRI.

(Remarks on code availability)

Version 3:

Reviewer comments:

Reviewer #1

(Remarks to the Author)

I thank the authors for providing evidence that MNase and Tn5 chromatin NRI values share similar profiles, which successfully addresses my concern that the MNase NRI might result from the A/T bias of MNase rather than reflecting chromatin characteristics. While the statement that 'higher NRI values are caused by tighter nucleosome wrapping' seems physically plausible, I remain concerned that other factors may contribute to the NRI values beyond the strength of nucleosome wrapping. Although I lack evidence to support this concern, and I understand it may be difficult to fully address in this manuscript, I hope future work will provide robust evidence to substantiate the claim that 'higher NRI values are caused by tighter nucleosome wrapping,' rather than relying on physically reasonable speculation. Lastly, I appreciate the authors' patience and effort in addressing my concerns throughout the review process.

(Remarks on code availability)

REVIEWER COMMENTS

Reviewer #1 (Remarks to the Author):

The authors used the MNase-X-ChIP-seq protocol to map the lengths of DNA fragments resulting from MNase enzyme cleavage across genome-wide. Utilizing this data, they introduced metrics, namely the Nucleosome Wrapping Score (NRS) and Nucleosome Wrapping Index (NRI), to describe the average wrapping states of nucleosomes within specific genomic regions. Upon analyzing the distribution of nucleosome wrapping states across genome-wide, the authors identified a very intriguing phenomenon: the genome can be partitioned into tightly wrapped nucleosome wrapping domains (TiNRDs) and loosely wrapped nucleosome wrapping domains (LoNRDs); TiNRDs and LoNRDs precisely coincide with A compartment domains and B compartment domains, respectively. I have the following concerns:

Major

1. The research community dedicated to investigating 3D chromatin organization broadly agrees that chromosomes are hierarchically organized into large A/B compartments, further divided into smaller topologically associating domains (TADs). The work here presented, claims to reveal a novel level of chromatin organization. However, it simultaneously emphasizes that TiNRDs and LoNRDs precisely coincide with widely studied A and B compartments, respectively. Thus, the clarity of the novel level of chromatin organization is needed.

Response: We thank the reviewer for bring this inaccurate description to our attention. We agree that nucleosome wrapping domain, which is not a 3D structural entity of chromatin or genome, does not represent “a novel level of chromatin organization”. What we hope the readers will be convinced is that we have discovered a novel domainization principle of the chromatin state encoded by nucleosome wrapping states. We modified the term “a novel level of chromatin organization” as “a previously unrecognized domainization principle of the chromatin” in the revised manuscript.

2. The work here presented, also claims to reveal a novel principle of chromatin organization. However, the investigation primarily focuses on the wrapping length of DNA on nucleosomes, uncovering a novel phenomenon or feature within euchromatin and heterochromatin – specifically, that the wrapping lengths of DNA on nucleosomes within euchromatin are likely longer than those within heterochromatin. Despite this revelation, the study does not establish a causal relationship between this feature and the formation of NRDs or compartments. Further clarification is necessary to elucidate the proposed novel principle of chromatin organization.

Response: We thank the reviewer for bring this inaccurate description to our attention again. We agree that 'chromatin organization' tends to refer to the three-dimensional or spatial structure of chromatin, especially in the field of 3D chromatin organization. The analogy of primary, secondary, and tertiary structures of proteins to is often used to describe the hierarchical structure features of

chromatin. In this article, we focused on the wrapping states of DNA on nucleosomes, which can be considered as a key aspect of chromatin's primary structure. In the field of epigenetics, "region" or "domain" are usually used to describe the structural or functional states of the primary structure of chromatin, e.g. differentially methylated regions, H3K27me3 domains. Therefore, we hope the reviewer will agree that "a previously unrecognized domainization principle of the chromatin" serves better than "a novel principle of chromatin organization" in this manuscript.

We preliminarily explored the mechanisms regulating the nucleosome wrapping state in this study. We found that transcription promotes the wrapping of nascent nucleosomes. However, how the nucleosome wrapping domains form and whether there is a causal relationship between nucleosome wrapping and Hi-C compartment domains or replication timing domains remains to be explored.

3. Fig. 3a displays a clear positive correlation between H3 NRI(140) and DNase I sensitivity, while Fig. 1d reveals a similar correlation between H3 NRI(140) and H3 ChIP signals. This indicates that chromatin regions with high DNase I sensitivity (indicating low nucleosome density) also exhibit high H3 ChIP signals (indicating high nucleosome density). To enhance the study's credibility, the author should provide an explanation for this apparent contradiction.

Response: We appreciate the reviewer for highlighting this crucial point, and we totally understand the concern. DNase I hypersensitive sites (DHSs) are short regions of chromatin that are highly sensitive to cleavage by DNase I. DHSs are characterized by free of nucleosome binding. To analyze the nucleosome signals around DHSs, we downloaded the DNase-seq bigwig signal file and peak file of mouse ES cell line E14 under the accession ENCSR000CMW from ENCODE (<https://www.encodeproject.org>). Then we calculate nucleosome distribution within the 1 Kb region around centers of DHSs. As shown in response Fig.1, H3, H4 or Input of our wrapping-seq data all show depleted signal at the center of the DHSs. When these data were viewed on a genome browser under high zoom in level (e.g. IGV tracks with 1 Kb bar), we can see that the selected DHSs are located at nucleosome free regions (response Fig.2). However, when we gradually zoom out (e.g. IGV tracks with 100 Kb or 1 Mb bar), we can see that the DHSs are not evenly distributed across the genome, with some genomic regions show clusters or high density of DHSs, while adjacent regions show sporadic or no DHS. The clusters of DHSs are coincident with high H3, H4 or Input signals. When viewing the whole chromosome 1 (e.g. IGV tracks with 10 Mb bar), it's evident that the domains of DNase I hypersensitive coincident with the domains of H3, H4 or Input signals. These observations are also supported by the genome-wide correlation analysis with various bin sizes. As shown in response Fig.3, at the 1 Kb regions around DHS peaks, the correlation between DNase I hypersensitivity and H3, H4 or Input signals of wrapping-seq is weak, especially for H3 and H4. However, when 10 Kb, 100 Kb and 1 Mb genomic bins were used, this correlation increased substantially. These results suggested that, on one hand, DHSs are usually featured by free of nucleosome; on the other hand, DHSs are highly clustered on large genome scale and coincident with high nucleosome density. The reason and biological significance of this correlation remains to be investigated.

Response Fig.1, heatmap shows the distribution of nucleosome signals around the centers of DHSs.

Response Fig.2, IGV tracks show the distribution of DNase I hypersensitivity signal and nucleosome signals on chromosome 1 with different zoom in levels.

Response Fig.3, Dot plots show the correlation between DNase I hypersensitivity signal and H3, H4 or Input signals of MNase-X-ChIP, in DNase I peak regions, 10 Kb, 100 Kb and 1 Mb genome bins. “r” indicates Pearson correlation coefficient.

4. In the section titled 'Transcription promotes nascent nucleosome wrapping,' the author labels nascent nucleosomes using EdU and subsequently conducts a wrapping-seq experiment to explore nucleosome wrapping dynamics during DNA replication. In this section, it is crucial for the author to initially rule out the influence of EdU labeling on the distribution of fragment length (as shown in Fig. 1a) and the calculation of NRI/NRS values. Without such exclusion, it becomes challenging to determine whether the observed differences in NRS values are attributed to DNA replication or EdU labeling.

Response: We thank the reviewer for this valuable advice. As suggested, we analyzed the fragment length of DNA of the input sample, which represents the steady-state chromatin, and the biotin

pulldown sample, which represents the nascent nucleosomes, under both “Pulse” and “Chase” conditions. We found that the fragment length profiles of steady-state nucleosomes and nascent nucleosomes show little difference, under both “Pulse” and “Chase” conditions (Response Fig.4). These results suggested that the incorporation of EdU into the nucleosome DNA did not change the digestion behavior of MNase on nucleosomes.

Response Fig.4, Meta profiles show the DNA fragment length of Input (parental nucleosomes) and Pulldown (nascent nucleosomes) samples, under both “Pulse” and “Chase” conditions, with DMSO or TPL treatment.

To further test the potential effects of EdU labeling on nucleosome wrapping, we extensively labeled the mES cells with EdU for 16 hours, then performed xMNase-wrapping-seq. We found that the genome-wide NRIs of labeled and non-labeled cells are the same (Response Fig.5, added as Fig. S4a), and highly correlated at genome-wide (Response Fig.6, added as Fig. S4b). These results suggest that EdU labeling does not have a significant effect on nucleosome wrapping states.

Response Fig.5 IGV tracks show the NRI(140)s of xMNase-wrapping-seq of two non-label replications and two EdU labeled replicates. The NRIs are calculated with 100 Kb bin resolution. This figure is added in the revised manuscript as Fig. S4a.

Response Fig.6, Dot plot shows the correlation between NRI of non-label cells and NRI of EdU-labeled cells. “r” indicates Pearson correlation coefficient. This figure is added in the revised manuscript as Fig. S4b.

In addition to these results, it’s worth noting that we observed increased nucleosome wrapping score (NRS) of the nascent nucleosomes from pulse to chase condition. The same trend is observed using published MINCE data (Ramachandran and Henikoff, 2016, doi: 10.1016/j.cell.2016.02.062). These results suggested that EdU labeling can be tolerated by the wrapping dynamics of nascent nucleosomes.

5. In Fig. 1e, certain chromatin regions exhibit a notable change in NRS under different digestion time, while NRI remains relatively constant. Consequently, I recommend that the author uses NRI instead of NRS as a metric for assessing nucleosome wrapping states in the section 'Transcription promotes nascent nucleosome wrapping' and in Fig. 4. Considering the author's claim that 'Input' (parental) and 'Pulldown' (nascent) nucleosomes in each 'Nascent' or 'Chase' condition undergo the same MNase digestion, it is expected that NRI values would yield similar results. (as NRS?)

Response: We thank the reviewer for this suggestion. We have initially shown the NRI results related to wrapping dynamics of nascent nucleosome in the original Fig. S4a, S4b and S4d (revised Fig. S4c, S4d and S4f). We apologize for inadequate explanation of NRS/NRI and the reason we choose NRS particular to compare the nucleosome wrapping dynamics after DNA replication. This concern is addressed as below, and we have rephrased the text of line 102-127 and line 290-303 in the revised manuscript accordingly.

To calculate the nucleosome wrapping score (NRS) for a genomic interval, we first counted the fragment length of all the DNA fragments mapped within that region. Then we separated the DNA fragment into two groups based on the fragment length. For example, if the “break point” is 140bp, then the nucleosome wrapping score is computed as the relative deviation between the number of DNA fragments within 140-250 bp (longer ones) and the number of DNA fragments within 50-140 bp (shorter ones). As shown by Fig. 1e and Fig. S1i, when the time of MNase digestion increases, the ratio of long DNA fragment decreases and the ratio of short DNA fragment increases, and the NRS decreases accordingly. Thus, NRS is sensitive to the variation of MNase digestion, and cannot be used for comparing nucleosome wrapping dynamics from two different MNase digestions. Nucleosome wrapping index (NRI) is a z-score transformation of NRSs of all genomic intervals. Thus, NRI represent the ordering of nucleosome wrapping degree of all the genomic intervals. As

shown by Fig. 1e and Fig. S1j, NRI is insensitive to the variation of MNase digestion. In other words, NRS is an absolute metric of wrapping degree, and NRI is a relative metric of wrapping degree.

As the reviewer suggested, we have initially calculated NRI for the Input and Pulldown nucleosomes for both “Pulse” and “Chase” conditions. However, we found that the genome-wide NRI is highly correlated and stably between the parental nucleosomes and nascent nucleosomes at genome-wide (Fig. S4c after revised) and in NRDs (Fig. S4d after revised). These results suggested that the genome-wide ordering of nucleosome wrapping degree is stably inherited after DNA replication, meaning that the more tightly wrapped nucleosomes are still wrapped more tightly, and more loosely wrapped nucleosomes are still more loosely wrapped after DNA replication, even if the whole picture of wrapping may have changed.

Thus, to compare the wrapping dynamics of nascent nucleosomes between “Pulse” and “Chase” conditions, we went back to use NRS with two reasons as explained below. First, as we pulse labeled the cells with EdU for 20 min, only a small portion of genome for each S-phase cell is labeled. Thus, the vast majority of 'Input' nucleosomes are parental nucleosomes with steady-state, and should have the same wrapping state when compared between “Pulse” and “Chase” samples. Second, in either “Pulse” or “Chase” condition, as the parental nucleosomes and nascent nucleosomes are within a same cell at the time of MNase digestion, they underwent exactly the same MNase digestion condition. Thus, the NRSs of parental nucleosomes and nascent nucleosomes can be compared directly within but not cross “Pulse” and “Chase” condition. Based on these two reasons, we used the parental nucleosomes as a “baseline” to analyze the dynamics of nascent nucleosomes. We observed that under “Pulse” condition, nascent nucleosomes have a smaller NRS than parental ones, but under “Chase” condition, their NRS surpasses that of the parental nucleosomes. The same trend is observed using published MINCE data (Ramachandran and Henikoff, 2016, doi: 10.1016/j.cell.2016.02.062). These results are shown as Figure 4b-4d and Figure S4d-S4g and described in the manuscript.

Taken together, we hope the reviewer is convinced that the dynamic wrapping of nascent nucleosomes can be compared via NRS but not NRI in the pulse-chase experiment.

6. In Fig. 4d, the author employs a bin size of 100kb for the NRS calculation, which is close to the size of DNA replication domains. It's worth noting that chromatin is typically partitioned into TADs with a median size of 185kb (as reported in Cell 159, 1665-1680), and TADs are recognized as stable units of replication timing regulation (as reported in Nature 515, 402-405, 2014). Given this context, I recommend that the bin size used to measure nucleosome wrapping states during the DNA replication process be much smaller than the size of replication domains.

Response: We thank the reviewer for this kind suggestion. TADs are megabase-scale contact domains, with a median size of 880 Kb in mouse ES cells (Nature 485, 376–380, 2012). subTADs are sub-megabase-scale chromatin domains nested hierarchically within TADs, with a median size of 185 kb (Cell 159, 1665-1680, 2014). As stated in Nature 515, 402-405, 2014, the size of ‘replication domains’ are within 400–800 kilobases. Thus, the 100 Kb bin size we used for the NRS

calculation is much smaller than the size of replication domains.

As suggested, we calculated the genome-wide NRS during DNA replication using bin size of 10 Kb, then we counted the NRS in the NRDs as did for Fig. 4d. As shown by Response Fig.7, the results are not different with those calculated using bin size of 100 Kb as shown in Fig. 4d. Thus, we will keep the results using 100 Kb bin in the revised manuscript.

Response Fig.7, Dot plots show the difference of NRS between parental and nascent nucleosomes as distance to the diagonal. TPL was used to inhibit transcription initiation, and DMSO treatment was control. NRS was calculated with 10 Kb bin size.

Minor

7. As shown in Figure S1e, the NRI values vary with respect to the size of each bin when computing the NRS values. It is necessary to specify the bin sizes of NRI values used in other figures in this work.

Response: We thank the reviewer for this kind suggestion. As most of the figures contains NRI or NRS, we added “The bin size of 100 kb will be used for most downstream analysis unless noted.” at line 136 of revised manuscript. We also added notes of bin size in figures where needed, such as Fig. S2b, Fig. 3d, 3e, 3f, Fig. S3c.

8. Does the “MNase input” in Fig.1d mean classical MNase-seq data or MNase-seq data from wrapping-seq? Please use some terms to distinguish them.

Response: We thank the reviewer for pointing out this issue. We will use “MNase-X-ChIP input” to refer to the MNase-seq signal from our histone-wrapping-seq, and to distinguish from the classical “MNase-seq” data. We have changed notes in revised Fig. 1d.

9. One type error in line 175 “Correspondingly, quantification of NRS in these peak regions showed that nucleosomes in H3K4me1 or H3K4me3 peaks regions have higher H3 NRS(140) that(should be than) those in H3K9me3 or H3K27me3 peaks regions (Fig. S3a).”

Response: We thank the reviewer for noticing this typo, we have corrected it in the revised manuscript.

Reviewer #2 (Remarks to the Author):

In this paper, the authors utilize a method they had previously developed termed MNase-X-ChIP-seq to characterize nucleosome wrapping genome wide. This pipeline, termed wrapping-seq, allows the authors to generate a nucleosome wrapping score (NRS) using the relative balance of fragment sizes at discrete genomic bins. This wrapping score is then converted to a z-score, which the authors term the nucleosome wrapping index (NRI). Using the NRI, the authors provide data suggesting that nucleosome wrapping varies significantly across the genome, with euchromatin regions showing much higher wrapping when compared to heterochromatic regions. The authors further demonstrate that newly replicated regions show lower levels of wrapping when compared to steady-chromatin, consistent with previous observations that nascent chromatin is hypersensitive to MNase digestion. Interestingly, the authors show that transcription plays an active role in promoting DNA wrapping, as inhibition of RNA pol II after replication fork passage prevents nascent chromatin from acquiring steady-state levels of nucleosome wrapping. Lastly, authors show that tightly-wrapped domains (TiNRD) and loosely-wrapped domains (LoNRDs) correlate with A and B compartments identified by Hi-C. These observations are interesting and reveal potentially important insights into differences in chromatin structure across distinct domains. However, there are several key issues that must be addressed that are described hereafter:

Major concerns:

1. No mention of MNase sequence bias. MNase has a known sequence bias, in that it digests AT-rich DNA preferentially over GC-rich DNA. Sequence composition varies significantly across the genome in a manner that could impact the NRI measurements calculated by the authors. Authors should discuss this bias and demonstrate that sequence biases do not strongly contribute to the observations made in this manuscript, particularly the strong NRI differences noted between euchromatin and heterochromatin.

Response: We appreciate the reviewer for highlighting this critical point, and we totally understand the concern. We performed extensive analysis between A/T content and fragment length or NRI. These results argue against that the A/T bias of MNase contribute significantly to the DNA fragment length profile or observed nucleosome wrapping states. We added the following text and related figures to the revised manuscript at line 215-237:

“It’s reported that MNase has a sequence bias towards A/T (Dingwall, C., et al. *Nucleic Acids Res* 9(12): 2659-2673; Hörz, W. and W. Altenburger. *Nucleic Acids Res* 9(12): 2643-2658). Consistently, when analyzing the DNA fragments of H3-wrapping-seq, we observed high A/T frequency at both upstream and downstream of the cutting sites of DNA fragments, mapped within either TiNRDs or LoNRDs (Response Fig.8, added as Fig. s2f). We also observed high G frequency downstream of A/T or high C frequency upstream of A/T, at the direction of top strand (Response Fig.8, added as Fig. s2f). Although the DNA fragments from TiNRDs have lower average A/T content than those from LoNRDs, they show no substantial difference in MNase cutting bias, suggesting that the sequence bias of MNase is not systematically biased towards TiNRDs or LoNRDs. As each DNA fragments from H3 wrapping-seq represents a single nucleosome cut out

from the chromatin, we analyzed the correlation between fragment length and the A/T content or dinucleotide frequency of A/TG and CA/T for each DNA fragment. We found that the correlations are weak for DNA fragments from either TiNRDs or LoNRDs (Response Fig.9, added as Fig. 2e, 2f), suggesting that the A/T content and dinucleotide frequency *per se* do not determine the DNA length protected by a nucleosome. To analyze the genome-wide correlation between the A/T content and H3 NRI at single nucleosome-level, we used 100 bp bin size to calculate and count the signals of A/T content and H3 NRI. We found that the Pearson correlation coefficient between A/T content and H3 NRI is -0.354 (Response Fig.10, added as Fig. 2g), indicating weak correlation. IGV track views supported this weak correlation. For example, regions “R1” and “R2” have comparable A/T content, but H3 NRI varies greatly; regions “R3” and “R4” have apparently different A/T content, but the H3 NRI values are similar (Response Fig.11, added as Fig. 2h). Taken together, we did not see a strong correlation between genome A/T content and DNA fragment length or nucleosome wrapping index, suggesting that the genome A/T content and MNase sequence bias did not contribute significantly to the nucleosome wrapping states we observed.”

Response Fig.8, Mete profiles show the base frequency around the 5’ or 3’ cut site of DNA fragments mapped within either TiNRDs or LoNRDs. Added in revised manuscript as Fig. s2f.

Response Fig.9, Dot plots show the correlation between fragment length and A/T content or AG/TG/CA/CT frequency of DNA fragments mapped within either TiNRDs or LoNRDs. “r” indicates Pearson correlation coefficient. Added in revised manuscript as Fig. 2e, 2f.

Response Fig.10, Dot plots show the correlation between A/T content and H3 NRI counted with 100 bp bin size. “r” indicates Pearson correlation coefficient. Added in revised manuscript as Fig. 2g.

Response Fig.11, IGV tracks show genome-wide distribution of A/T content and H3 NRI counted with 100 bp bin size. “R1”, “R2”, “R3”, “R4” indicate four selected genome intervals. Added in revised manuscript as Fig. 2h.

In addition, results from Figure 4 and 5 also support that the nucleosome wrapping states we observed are not artificial. In figure 4, based on the analysis of nucleosome wrapping score, we observed that nascent nucleosomes wrap more loosely than parental ones right after assembly, but wrap more tightly than parental ones after maturation for 1 hr (Fig. 4b, 4c, 4d). The same trend is also observed from published MINCE-seq dataset from another lab (Fig. S4c). Moreover, the wrapping dynamics can be slowed down when inhibiting transcription (Fig. 4d). In figure 5, we observed that NRI is highly correlated with replication timing value and Hi-C PC1 value, which are derived by totally different experimental methods that do not involve MNase digestion.

Taken together, we hope the reviewer are convinced that the nucleosome wrapping states we observed are not artificial, and that the A/T content of genome and A/T sequence bias of MNase did not contribute significantly to results we observed in this study.

2. The authors claim that because NRI patterns from different digestion time points are highly similar, MNase-seq NRI(140) is robust for nucleosome wrapping detection, despite of a certain degree of digestion variation. However, their time-course is not sufficient to be able to make these claims. Previous studies have shown significant variation in nucleosome recovery from 1-10 minutes of MNase treatment². Indeed, many of the most unwrapped, fragile nucleosomes, could be depleted from libraries after 10 minutes of treatment^{2,3}. Authors should evaluate NRI140 after 1 and 5 minutes MNase digestion to fully validate claims that: "NRI140 is robust for nucleosome wrapping detection."

Response: We thank the reviewer for bring this issue to our attention. We'd like to explain that our current research was focused on the wrapping states of nucleosomes across the entire genome, with both experimental conditions and analytical pipeline aiming for sequencing data of single-nucleosome that can cover the whole genome. Thus, when we intended to test whether a certain degree of variation of MNase digestion is acceptable for wrapping-seq, we choose MNase digestion times when mono-nucleosome was the dominant digestion product. As shown in the agarose gel (Response Fig.12, added as Fig. S1h), we can see that at 1 minute or 5 minutes, the proportion of single nucleosome DNA is low, and the genome coverage cannot be warranted. Our results show that varying the MNase digestion time within the range of 10-50 minutes does not significantly affect the NRI pattern. This result suggests that a certain degree of variation of MNase digestion can be tolerated by NRI. We have reorganized the text of line 141-150 in the paper as following to reflect this point and the limitation of this results:

“To further test whether MNase NRI is sensitive to variation of MNase digestion, we prepared xMNase-wrapping-seq libraries within the digestion time of 10~50 min, when mono-nucleosome was the dominant digestion product (Fig. S1h) ..., suggesting that MNase NRI is not sensitive to MNase digestion variation, as long as mono-nucleosome is the dominant product.”

On the other hand, as the reviewer mentioned, according to other reports (Yu, J., et al. Cell Rep 32(4): 107953.; Chereji, R. V., et al. Genome Biol 20(1): 198.), extremely short digestion only released nucleosomes from open chromatin regions, which is applicable for studying the wrapping states of the most unwrapped, fragile nucleosomes in open chromatin regions. While interesting, it

is beyond the scope of the current study.

3. The unusual correlation of NRI values to ChIP yield is strange, as it suggests that some aspect of the authors' workflow is losing signal in heterochromatic areas. The authors do a nice job addressing a potential cell cycle-related explanation, but the issue remains unresolved. One concern is that the heterochromatic regions are resistant to both MNase and sonication³. Authors should show that their MNase treatment conditions are able to sufficiently fragment and solubilize genomic DNA to allow for a fair comparison between euchromatin and heterochromatin.

Response: We thank the reviewer for raising this critical concern. We apologize for inadequate introduction of background for the questions we wanted to address and the rationale related to the cell cycle experiment (Fig. 1g and revised Fig. S11). We have re-written the text as following at line 153-186:

“We observed that the nucleosome density represented by H3 or H4 MNase-X-ChIP signals or MNase-X-ChIP input signal show obvious variation across the genome (Fig. 1d). Surprisingly, we observed that NRI is highly positively correlated with nucleosome density (Fig. 1d, Fig. 1f). As higher nucleosome density might provide more protection of MNase digestion, resulting in higher NRI value, we intended to test whether the NRI pattern still persists when there should be no variation of nucleosome density across the genome. However, before that, we also need to figure out whether the variation of nucleosome density is due to experimental error of wrapping-seq or due to unsynchronized DNA replication in a cell population. To this end, we sorted cells by G1, S and G2/M phases, and then performed xMNase-wrapping-seq using crosslinked chromatin, and genome sequencing using genomic DNA (“gDNA”) in parallel. We found that in all three cell cycle phases, the signal of genomic DNA and MNase digested chromatin is highly similar viewed from IGV (Fig. 1g) and highly correlated at genome-wide (Fig. S11). As the genomic DNA libraries represent genuine genome coverage, these results suggested that wrapping-seq *per se* did not introduce bias of nucleosome coverage of the genome. Moreover, we found that both chromatin and genomic DNA signal show more variation along the genome of S-phase cells than that of G1 or G2/M phase cells (Fig. 1g). To quantify the variation, we counted the genomic DNA and chromatin signals in 100 Kb genomic intervals, and show the distribution of signal as histograms. As shown in Response Fig.12 (revised Fig. 1h and 1i), S phase cells show a bimodal distribution of both genomic DNA and chromatin signals, whereas G1 and G2/M cells show a unimodal distribution of those signals. Moreover, the peak width of S phase cell signals is wider than that of G1 and G2/M cells. These results supported that the genomic DNA and chromatin signals show more variation in S phase cells than in G1 and G2/M cells. However, when we calculated the NRI pattern using the chromatin signal, we found that the NRI patterns of G1, S and G2/M phase cells are highly similar (Fig. 1g), and are equally well correlated with the MNase NRI of unsynchronized cells (Response Fig.13, revised Fig. S11). Thus, these results argue that the NRI pattern is not an artificial consequence of nucleosome density variation along the genome.”

For Fig. 1g specially, we noted that the genome-wide variation of genomic DNA and chromatin signals of S phase cells is not so easily captured in the original figure, as the track height is not enough. Thus, we adjusted the ranges of “chromatin” tracks and “gDNA” (genomic DNA) tracks

from the original “[0, 0.2]” to “[0.06, 0.12]” and “[0.07, 0.15]”, respectively, to make this variation of signal easier to spot.

Response Fig.12, Histograms show the distribution of reads density of MNase digested chromatin (Left panel) or sonicated genomic DNA (right panel) of G1, S and G2 phase cells. Added in the revised manuscript as Fig. 1h, 1i.

Response Fig.13, Histograms show the distribution of reads density of MNase digested chromatin (Left panel) or sonicated genomic DNA (right panel) of G1, S and G2 phase cells. Added in the revised manuscript as Fig. S1m.

4. The manuscript should be carefully proofread and edited for clarity and proper grammar.

Response: We thank the reviewer for this kind suggestion, we have edited some text and grammar in the article as marked in the revised manuscript.

Minor points:

(1) In figure S1K the G2-phase data is inappropriately labeled “S-phase”

Response: We thank the reviewer for noticing this error, we have corrected accordingly.

References:

1 Dingwall, C., Lomonosoff, G. P. & Laskey, R. A. High sequence specificity of micrococcal nuclease. *Nucleic Acids Research* 9, 2659-2674, doi:10.1093/nar/9.12.2659 (1981).

2 Chereji, R. V., Bryson, T. D. & Henikoff, S. Quantitative MNase-seq accurately maps nucleosome occupancy levels. *Genome Biology* 20, 198, doi:10.1186/s13059-019-1815-z (2019).

3 Mieczkowski, J. et al. MNase titration reveals differences between nucleosome occupancy and chromatin accessibility. *Nature Communications* 7, 11485, doi:10.1038/ncomms11485 (2016).

Reviewer #3 (Remarks to the Author):

The authors use MNase-digested DNA fragment lengths as a genome-wide measure of DNA unwrapping: Smaller fragments mean more unwrapping. This is seen across large swaths of eukaryotic genomes. More wrapping (longer DNA fragments) is seen in gene bodies versus elsewhere. Certain histone modifications and Pol II also track with gene bodies, and so the same results are seen where they are enriched. Unfortunately, essentially all the key results can be explained by the well-known A/T-sequence bias of MNase and the nonrandom distribution of A/T and G/C across eukaryotic genomes. The authors completely ignore this. Even the author's prior work in this area (Ref. 15) ignores this reality. This must be addressed definitively before this work can be meaningfully considered for publication.

Specific comments

L. 101-122: NRS is a metric of wrapping (and NRI is a statistical derivative, Z-score). However, this initial explanation in the beginning of the results is largely opaque to the reader. I had to go through the calculation simply to understand what was being done, but then came away with a metric that, at least to me, had uncertain meaning. The reader would be better served by having some text-based meaning of what, at the end of all this, is actually being measured. It appears to be there at the end of the first paragraph, and so it would be more helpful to move this up to the beginning, as a rationale as to why one chooses such a metric, and its caveats.

Response: We thank the reviewer for bringing this issue to our attention. We have restructured the text in the revised manuscript from line 102 to 112 as below, to strengthen the explanation of NRS and NRI and the comparison between them. We also added mathematical formulas for the z-score conversion from NRS to NRI in the methods section from line 484 to 487.

“As shown in Fig. 1a, to calculate the NRS (nucleosome wrapping score) for a genomic interval (e.g., a 10 kb bin), we first counted the length of all the DNA fragments mapped within that interval. Then we separated the DNA fragment into two groups by a fragment length “break point (bcp)” X bp. NRS is computed as the relative deviation between the number of fragments within X-250 bp (represented by variable “a”) and the number of fragments within 50-X bp (represented by variable “b”), expressed as $NRS(X) = (a-b)/(a+b)$. For example, if the “break point” is 140bp, then NRS(140) is computed as the relative deviation between the number of DNA fragments within 140-250 bp (longer ones) and the number of DNA fragments within 50-140 bp (shorter ones). Thus, larger NRS value indicates DNA wraps tighter on nucleosomes; smaller NRS value indicates DNA wraps looser on nucleosomes. It's reasonable that when the break point increased from 80 bp to 160 bp with 10

bp step, the genome-wide NRS generally shifted from 1 to -1 (Fig. 1b, Fig. S1b). However, when the raw NRS of each break point was transformed as Z-score to derive NRI (nucleosome wrapping index), the genome-wide NRIs of different break points show highly similar pattern (Fig. 1c, Fig. S1c) and high correlation (Fig. S1d). Thus, Z-score transformation eliminates the bias of nucleosome wrapping degree introduced by selecting the break point arbitrarily. Similarly, larger NRI value indicates DNA wraps tighter on nucleosomes; smaller NRI value indicates DNA wraps looser on nucleosomes. While NRS is an absolute metric of wrapping degree, NRI is a relative metric of wrapping degree, represent the ordering of nucleosome wrapping degree of all the genomic intervals.”

L123: “Bin lengths” is unclear. I think “genomic intervals (e.g., 1 kb, 10 kb, etc.)” would be clearer.

Response: We than the reviewer for this kind suggestion. We used “genomic interval” to explain the concept of NRS and NRI, and use “bin” or “bin size” when needed as they are a commonly used bioinformatic term. We also used one to label the other when necessary, such as in line 103 and line 133.

Fig 1f. Shouldn't the axis be flipped (and same with all other equivalent plots), with ChIP being the dependent variable? Also, I was not convinced that the positive correlation of NRI with H3 ChIP signal was due to differences in H3 density, as opposed to technical differences (e.g., in chromatin extraction, ChIP efficiency, library construction, PCR, and depth of sequencing). This has relevancy implications for Fig. 1g/S1K, making it moot. Fig. 1g is not interpretable by a reader. Fig. S1K has a typo (S phase). The conclusions from both are not well grounded. I'm not sure that the proposed problem that Fig. 1g was meant to address is even valid. Even in S phase the vast majority of the genome has a normal density of nucleosomes. So I don't see how this is a concern.

Response: We than the reviewer for raising these important questions, which are addressed separately as below.

1. We flipped the axis such that the independent variable is on the X-axis, and the dependent variable is on the Y-axis, where there is a potential causal relationship between the two variables, such as in Fig. 1f, Fig. 3a, Fig. 3c. For other dot plots, such as Fig. 4d and dot plots in supplemental figure, where the axis are intended for correlation but not for causal relationship interpretation, we kept the original ones.

2. We apologize for inadequate introduction of background for the questions we wanted to address and the rationale related to the cell cycle experiment (Fig. 1g and revised Fig. S11). We have re-write the text as following at line 153-187:

“We observed that the nucleosome density represented by H3 or H4 MNase-X-ChIP signals or MNase-X-ChIP input signal show very obvious variation across the genome (Fig. 1d). Surprisingly, we observed that NRI is highly positively correlated with nucleosome density (Fig. 1d, Fig. 1f). As higher nucleosome density might provide more protection of MNase digestion, resulting in higher

NRI value, we intended to test whether the NRI pattern still persist when there should be no variation of nucleosome density across the genome. However, before that, we also need to figure out whether the variation of nucleosome density is due to experimental error of wrapping-seq or due to unsynchronized DNA replication in a cell population. To this end, we sorted cells by G1, S and G2/M phases, and then performed xMNase-wrapping-seq using crosslinked chromatin, and genome sequencing using genomic DNA ("gDNA") in parallel. We found that in all three cell cycle phases, the signal of genomic DNA and MNase digested chromatin is highly similar viewed from IGV (Fig. 1g) and highly correlated at genome-wide (Fig. S11). As the genomic DNA libraries represent genuine genome coverage, these results suggested that wrapping-seq *per se* did not introduce bias of nucleosome coverage of the genome. Moreover, we found that both chromatin and genomic DNA signal show more variation along the genome of S-phase cells than that of G1 or G2/M phase cells (Fig. 1g). To quantify the variation, we counted the genomic DNA and chromatin signals in 100 Kb genomic intervals, and show the distribution of signal as histograms. As shown in Fig. 1h and 1i (Response Fig.12), S phase cells show a bimodal distribution of both genomic DNA and chromatin signals, whereas G1 and G2/M cells show a unimodal distribution of those signals. Moreover, the peak width of S phase cell signals is wider than that of G1 and G2/M cells. These results supported that the genomic DNA and chromatin signals shows more variation in S phase cells than in G1 and G2/M cells. However, when we calculated the NRI pattern using the chromatin signal, we found that the NRI patterns of G1, S and G2/M phase cells are highly similar (Fig. 1g), and are equally well correlated with the MNase NRI of unsynchronized cells (Response Fig.13, revised Fig. S1m). Thus, these results argue that the NRI pattern is not an artificial consequence of nucleosome density variation along the genome."

For Fig. 1g specially, we noted that the genome-wide variation of genomic DNA and chromatin signals of S phase cells is not so easily captured in the original figure, as the track height is not enough. Thus, we adjusted the ranges of "chromatin" tracks and "gDNA" (genomic DNA) tracks from the original "[0, 0.2]" to "[0.06, 0.12]" and "[0.07, 0.15]", respectively, to make this variation of signal easier to spot.

Response Fig.12, Histograms show the distribution of reads density of MNase digested chromatin (Left panel) or sonicated genomic DNA (right panel) of G1, S and G2 phase cells. Added in the revised manuscript as Fig. 1h, 1i.

Response Fig.13, Histograms show the distribution of reads density of MNase digested chromatin (Left panel) or sonicated genomic DNA (right panel) of G1, S and G2 phase cells. Added in the revised manuscript as Fig. S1m.

L156. At this point the author draw the conclusion of what NRS/NRI is measuring is in fact robust. It may be robust, but I am not convinced that the A/T-rich bias of MNase isn't driving the patterning. The authors should also plot A+T density, along with the NRI, and provide convincing data and analyses that rule out sequence bias of MNase digestion.

L175: mES have high GC-content in gene bodies making them more MNase resistant (high NRS) relative to intergenic regions. Yeast have high A/T-content in intergenic regions making the gene bodies relatively more MNase resistant. Thus, the correspondence between yeast and mouse is not surprising, and completely explainable by DNA sequence.

L182+: Since histone modifications and Pol II occupancy follow gene bodies, then MNase sequence bias explains these results as well.

Response: Specific comments of “L156”, “L175” and “L182+” are all rooted in the concern of MNase sequence bias, and they are addressed together as following.

We appreciate the reviewer for highlighting this critical point, and we totally understand the concern. We performed extensive analysis between A/T content and fragment length or NRI. The results argue against that the A/T bias of MNase contribute significantly to the DNA fragment length profile and nucleosome wrapping states we observed. We added the following text and related figures to the revised manuscript at line 215-237:

“It’s reported that MNase has a sequence bias towards A/T (Dingwall, C., et al. *Nucleic Acids Res* 9(12): 2659-2673; Hörz, W. and W. Altenburger. *Nucleic Acids Res* 9(12): 2643-2658). Consistently, when analyzing the DNA fragments of H3-wrapping-seq, we observed high A/T frequency at both upstream and downstream of the cutting sites of DNA fragments, mapped within either TiNRDs or LoNRDs (Response Fig.8, added as Fig. s2f). We also observed high G frequency downstream of A/T or high C frequency upstream of A/T, at the direction of top strand (Response Fig.8, added as Fig. s2f). Although the DNA fragments from TiNRDs have lower average A/T content than those from LoNRDs, they show no substantial difference in MNase cutting bias, suggesting that the sequence bias of MNase is not systematically biased towards TiNRDs or LoNRDs. As each DNA fragments from H3 wrapping-seq represents a single nucleosome cut out from the chromatin, we analyzed the correlation between fragment length and the A/T content or dinucleotide frequency of A/TG and CA/T for each DNA fragment. We found that the correlations are weak for DNA

fragments from either TiNRDs or LoNRDs (Response Fig.9, added as Fig. 2e, 2f), suggesting that the A/T content and dinucleotide frequency *per se* do not determine the DNA length protected by a nucleosome. To analyze the genome-wide correlation between the A/T content and H3 NRI at single nucleosome-level, we used 100 bp bin size to calculate and count the signals of A/T content and H3 NRI. We found that the Pearson correlation coefficient between A/T content and H3 NRI is -0.354 (Response Fig.10, added as Fig. 2g), indicating weak correlation. IGV track views supported this weak correlation. For example, regions “R1” and “R2” have comparable A/T content, but H3 NRI varies greatly; regions “R3” and “R4” have apparently different A/T content, but the H3 NRI values are similar (Response Fig.11, added as Fig. 2h). Taken together, we did not see a strong correlation between genome A/T content and DNA fragment length or nucleosome wrapping index, suggesting that the genome A/T bias and MNase cutting bias did not contribute significantly to the nucleosome wrapping states we observed.”

Response Fig.8, Mete profiles show the base frequency around the 5’ or 3’ cut site of DNA fragments mapped within either TiNRDs or LoNRDs. Added in revised manuscript as Fig. s2f.

Response Fig.9, Dot plots show the correlation between fragment length and A/T content or AG/TG/CA/CT frequency of DNA fragments mapped within either TiNRDs or LoNRDs. “r” indicates Pearson correlation coefficient. Added in revised manuscript as Fig. 2e, 2f.

Response Fig.10, Dot plots show the correlation between A/T content and H3 NRI counted with 100 bp bin size. “r” indicates Pearson correlation coefficient. Added in revised manuscript as Fig. 2g.

Response Fig.11, IGV tracks genome-wide distribution of A/T content and H3 NRI counted with 100 bp bin size. “R1”, “R2”, “R3”, “R4” indicate four selected genome intervals. Added in revised manuscript as Fig. 2h.

In addition, results from Figure 4 and 5 also support that the nucleosome wrapping states we observed are not artificial. In figure 4, based on the analysis of nucleosome wrapping score, we observed that nascent nucleosomes wrap more loosely than parental ones right after assembly, but wrap more tightly than parental ones after maturation for 1 hr (Fig. 4b, 4c, 4d). The same trend is also observed from published MINCE-seq dataset from another lab (Fig. S4c). Moreover, the wrapping dynamics can be slowed down when inhibiting transcription (Fig. 4d). In figure 5, we observed that NRI is highly correlated with replication timing value and Hi-C PC1 value, which are derived by totally different experimental methods that do not involve MNase digestion.

Taken together, we hope the reviewer are convinced that the nucleosome wrapping states we observed are not artificial, and that the A/T content of genome and A/T sequence bias of MNase did not contribute significantly to results we observed in this study.

L217: Subtitle is inappropriate for the paragraph that follows.

Response: We appreciate the reviewer for highlighting this point. However, after double checking of line marks such as “L156”, “L175”, “L422”, we found that the line marks we have maybe different from those the reviewer saw, so we are not sure which subtitle the reviewer refers to.

L422: There is no description of how the NRI was calculated. All I could find was “Then the genome-wide NRS was transformed as z-score to derive genome-wide nucleosome wrapping index (NRI).” Therefore no independent assessment of what NRI is measuring could be determined.

Response: We thank the reviewer for bringing this issue to our attention. We have added mathematical formulas of the z-score conversion from NRS to NRI in the methods section from line 484 to 487.

$$Z(i) = \frac{X(i) - \mu}{\sigma}$$

Z(i) is the Z-score of the NRS of the i-th genomic interval, x(i) is the NRS of the i-th genomic interval, μ is the mean of NRS of all genomic intervals, and σ is the standard deviation of NRS of all genomic intervals.

REVIEWER COMMENTS

Reviewer #1 (Remarks to the Author):

I would like to thank the authors for their response to my concerns. However, I still have one final query regarding the interpretation of the output of the MNase-X-ChIP-seq experiment. In the manuscript, the authors state that they use the MNase enzyme to cleave DNA and utilise the lengths of DNA fragments to calculate NRS/NRI. The larger NRI values observed in H3-wrapping-seq suggest that DNA wraps more tightly on nucleosomes. This claim is based on the supposition that DNA is stably wrapped on nucleosomes and that DNA fragments are protected by histones from MNase cleavage. However, as stated in lines 45-47, DNA located on nucleosomes can rapidly transition between an "unwrapping" and "rewrapping" state. It is therefore reasonable to speculate that in the MNase enzyme digestion process, DNA fragments may also transition from histone wrapping state to histone unwrapping state. If the aforementioned speculation is indeed accurate, it can be argued that DNA regions with a higher NRI value indicate that histone proteins exhibit a preference for long DNA fragments from this region. It would be beneficial to ascertain whether there is any evidence to disprove this later speculation.

Response: We appreciate the reviewer for highlighting this crucial point.

In this study, we aimed to analyze the wrapping states of DNA on nucleosomes by using the length of nucleosome-protected DNA fragments after MNase digestion. This principle has been used in other studies to analyze subnucleosomes (Ramachandran, S., et al. (2017) Mol Cell 68(6): 1038-1053.e1034). DNA located at the entrance and exit sites of nucleosomes can rapidly transit between "unwrapping" and "rewrapping" states. However, this process will not lead to spontaneous disassembly of the nucleosome. For example, it was estimated based on the FRET efficiency that 7–8 bp terminal DNA may stretched out in the open or unwrapping state (Wei, S., et al. (2015). Nucleic Acids Res 43(17): e111). Considering the inherent dynamics of nucleosomes, we strengthened the specificity of detection through two experimental conditions: 1. By fixing the interaction between DNA and histones via formaldehyde crosslinking, thereby reducing the spontaneous un-wrapping and re-wrapping of DNA on nucleosomes; 2. By fully digesting to remove base pairs unwrapping from the nucleosome core, thereby reflecting the length of the nucleosome-protected DNA. Thus, "histone proteins exhibit a preference for long DNA fragments" may still mean that histone proteins bind and protected longer DNA fragments, suggesting that nucleosomes in these regions have more tightly or fully wrapped states.

However, we do understand that we can only see one dimension of the genuine wrapping states of nucleosomes *in vivo* through wrapping-seq, and there is still uncertainty in interpretation of the biochemically basis of "the output of the MNase-X-ChIP-seq experiment" and NRI. Other technics, such as single-molecule and structural techniques, are needed to depict the full dimensions of the dynamics of nucleosomes *in vivo*. This limitation of wrapping-seq is now

pointed out in the discussion.

Reviewer #2 (Remarks to the Author):

In this revised version of the manuscript, the authors have satisfactorily addressed the major concerns. The authors should further proofread the manuscript to address remaining grammatical errors.

Response: We appreciate the reviewer's acknowledgement of our previous response and suggestion of further proofreading.

Reviewer #3 (Remarks to the Author):

I appreciate the reviewers attempt to address the reviewer's concerns and revising the manuscript. The manuscript reads more clearly, and allows us to examine it more thoroughly. The biggest sticking point was that the results could be explained by the A/T-bias of MNase. The new analysis does not change this, and in fact seems to provide evidence that the results can indeed be explained by MNase sequence bias.

Main comments on A/T bias

1. The authors find that there is high A/T content at cleavage sites (Fig. S2f), which only reconfirms the intrinsic sequence bias known for MNase. As the author point out, the A/T bias at the cleavage site is the same for Lo vs Ti NRDs. This is expected because at the cleavage site, one is simply measuring the A/T bias that is intrinsic to cleavage by MNase. It is not measuring the propensity to form Lo vs Ti NRDs. This can be found in the flanking genomic sequences that are not protected. Indeed, they find high A/T content in the flanking regions of LoNRDs compared to TiNRD, which is what one would expect if the LoNRD vs TiNRD pattern is driven by A/T bias. Since the flanking region outside of the protected region is more sensitive to MNase, one expects more cleavage opportunities and thus a higher proportion of LoNRDs over TiNRDs. One can discuss this back and forth, but the real answer resides in a control experiment, which has not been done. For example, if naked DNA is digested with MNase to the same size range, what would the NRI map look like when analyzed side-by-side with chromatin

Response: We thank the reviewer for bringing this issue to our attention again. To address this concern, we believe that “the A/T bias of MNase” and “the role of A/T content in nucleosome wrapping” should be considered separately.

Regarding the A/T bias of MNase, based on the results in Fig. S2f, we agree that MNase indeed showed A/T preference when digesting DNA outside of nucleosomes.

Concerning the role of A/T content in nucleosome wrapping, we agree that the genomic A/T

content might play a role in regulating nucleosome wrapping, but the specific mechanism requires further investigation. On one hand, the results from Fig. 2e indicate that, although DNA fragments with the highest A/T content tend to be shorter than those with lowest A/T content, there is significant variability in the A/T content within any given range of fragment length. This finding suggests that the length of nucleosome protected DNA fragments is not determined by A/T content alone. On the other hand, we did observe a negative correlation between A/T content and NRI ($r = -0.35$) (Fig. 2g). It's reported that a higher force is needed to unwrap the more flexible DNA side of a nucleosome than the less flexible side, when stretched by an optical tweezer (Ngo, T. T. M., et al., 2015, Cell 160(6): 1135-1144), and that TA dinucleotides are critical for accommodating the flexibility of nucleosomal DNA (Chua, E. Y., et al., 2012, Nucleic Acids Res 40(13): 6338-6352). Thus, A/T content may regulate nucleosome wrapping through multiple mechanisms.

We thank the reviewer for the suggestion of using MNase to digest genomic DNA as a control. However, based on that the rate of cleavage by MNase is 30 times greater at the 5' side of A or T than at G or C (Dingwall, C., et al. (1981). Nucleic Acids Res 9(12): 2659-2673), the DNA fragments from A/T-rich regions would be systematically shorter than those from G/C-rich genomic regions. Therefore, we believe that *in vitro* reconstituted chromatin would be a better method to study the factors regulating nucleosome wrapping, including both DNA sequence and epigenetic regulators, such as chromatin remodelers.

2. The analysis in Fig 2e, 2f does not seem appropriate for addressing the MNase bias because the authors are looking at the A/T sequence content of protected fragments, rather than the presumably unprotected DNA located immediately adjacent. Also, it is not clear how plotting fragment length vs A/T frequency addresses the question. Protected fragments regardless of length should have the same A/T content, otherwise MNase would preferentially digest the A/T-rich ones, thereby depleting A/T from the population.

Response: We appreciate the reviewer for raising this concern. We agree that analysis in Fig 2e, 2f can not address the question of MNase bias. Actually, we were trying to explore whether the A/T content of DNA fragments contribute to length of protected DNA fragments. We found that the A/T content of these fragments can range from as high as 75% to as low as 25%. Nonetheless, some of them with very different A/T content can have the same range of fragment length, meaning that the length of nucleosome protected DNA fragments is not determined by A/T content.

3. Fig. 4g indicates that higher NRI (more "fully wrapped") nucleosomes have low A/T content. The correlation is -0.35. I think it is arbitrary as to whether one sees this correlation as being weak or not weak. Nonetheless, this trend indicates that an A/T bias of MNase cannot be excluded as a likely explanation for all the results.

Response: We thank the reviewer for pointing out this issue. The Pearson correlation coefficients are interpreted following the rule of thumb guide. However, we do agree that the genomic A/T content might play a role in regulating NRI, but the specific mechanism requires further investigation.

4. The relationship of “tighter wrapping” with transcription is consistent with transcription occurring at CpG islands in mammals which are inherently more resistant to MNase (due to A/T cleavage bias) and thus will have their “wrap” skewed to longer fragment lengths. The authors should conduct the same analyses on their *Drosophila* and Yeast data, which do not have CpG islands.

Response: We thank the reviewer for the suggestion. CpG islands (CGIs) are, on average, 1000 base pairs (bp) long. The mouse genome contains 23021 CGIs, about half of which coincide with gene promoters (Deaton, A. M. and A. Bird (2011). *Genes Dev* 25(10): 1010-1022.). In our manuscript, the relationship between nucleosome wrapping and transcription activity is shown by Fig. 3c and 3d. In Fig. 3c, the units for counting signals of NRI and Polr2a are the whole gene bodies, but not just promoter regions. As shown by Fig. 3d, it's the NRIs of gene bodies, but not of promoters, decreases more significantly along with the decrease of RNAPII. Thus, the positive correlation between nucleosome wrapping and transcription activity is based on analysis of whole gene bodies of all genes, contain CGIs or not, but not only CGIs.

Other major comments

5. One would need to control for variable chromatin/DNA extraction efficiency from cells. This could be examined with the whole genome sequencing data generated for Fig. 1h. The authors note that NRI correlates with histone H3 ChIP signal. Could this mean that Lo NRD portions of the genome (B compartments?) extract less efficiently (possibly due to crosslinking)? If so, then small fragments might extract relatively more efficiently in such regions, which would leave larger fragments behind.

Response: We appreciate the reviewer for highlighting this point. As also suggested by the reviewer, we had already addressed the concern of chromatin or DNA extraction efficiency of wrapping-seq in the cell cycle experiment (Fig. 1g, Fig, S11). In that experiment, we had sorted cells by G1, S and G2/M phases, and then performed xMNase-wrapping-seq using crosslinked chromatin, and genome sequencing using genomic DNA (“gDNA”) in parallel. We found that in all three cell cycle phases, the signals of genomic DNA and MNase digested chromatin is highly similar viewed from IGV (Fig. 1g) and highly correlated at genome-wide (Fig. S11). As the genomic DNA libraries represent genuine genome coverage, these results suggested that the procedure of wrapping-seq did not introduce bias of nucleosome coverage of the genome.

6. Browser plots of NRI vs genomic coordinate along the x-axis (eg. Fig. 1c and more) lack a definitive y-axis range, such that one cannot evaluate the NRI values (Z-scores). It seems to me that the Z-scores barely reach 1, which means that across large genomes, these events will occur frequently by random chance. Fig. S2a-c reveals that the average NRI for an NRD bin is less than 0.5, which if I understand its meaning, is statistically not significant. I also wonder how smoothing affects correlations (Methods: “10 bins smoothed 100 kb bin NRI (140bp) data”). For example, a correlation scatter plot will inappropriately have a better correlation if the data are smoothed. Further, it is not clear how the values are tallied in browser windows (e.g., Fig. 1c). What happens when the quantity of values exceeds the number of pixels on the graph? Are they averaged? When reporting the number of LoNRD

vs TiNRD as in Fig 2b, statistics need to be reported, such as a violin plot of NRI values. This exists somewhat in Fig. S2a-c, but there is no discussion of how meaningful an average NRI (Z-score) of ~0.5 is.

Response: We appreciate the reviewer for raising these questions.

For browser plots, including Figures 1b, 1c, 1e, 1g, 2a, 2c, 2d, S1e, S4a, the signal ranges of NRS and NRI are indicated at the top of figures. We have added related notes in the figure legends. For the y-axis of these browser plots, if the value of a bin exceeded the upper range, it was automatically adjusted to the upper range; and similarly, if the bin value fell below the lower range, it was set to the lower range.

The distribution of all nucleosome wrapping index (Z-scores) are as shown in Fig. S1c, which should have a mean of 0 and a standard deviation of 1, as a normal distribution does. The browser plots, such as Fig. 1c and 1d, show that chromosome-wide NRI forms distinct domains, but not random distribution.

In Fig. S2a-c, the NRI of each NRDs is calculated as the average NRI of all bins contained within each NRD. As we can see from Fig. S1e, even within a LoNRD (domain with roughly continuous negative NRI), there will be bins with positive NRI, especially when seen with small bin such as 1 Kb bin; and *vice versa* for TiNRD. Thus, the range of the NRIs of NRDs is narrower than that of the NRIs of genome-wide bins.

The 10 bins smoothed 100 kb bin NRI (140bp) data of H3 was used for visualization in genome browser and NRD detection as recommended for replication timing domain detection (Ryba, T., et al. (2011). Nat Protoc 6(6): 870-895). Un-smoothed NRIs were used for annotation of genomic features. We have added notes in the method section (line 471).

As the main concerns of this comment is related to the data range of NRS/NRI, we want to refer to the reviewer to the calculation of NRS, which is computed as the relative deviation between the number (a) of relative longer DNA fragments and the number (b) of relative shorter DNA fragments, expressed as $(a-b)/(a+b)$. The NRS is intrinsically ranged from -1 to 1. Comparing with formula of a/b , $(a-b)/(a+b)$ is useful to avoid extreme values where a or b is extremely small. It's worth noting that even though the magnitude of variation of NRI might not very big along the genome, but we emphasize on the unexpected trend of roughly continuous distribution of negative or positive NRI along the genome, which we termed as "a previously unrecognized domainization principle of the chromatin" in the manuscript. However, we do understand that NRS/NRI is a relative but not absolute metric of nucleosome wrapping, and the interpretation of the biological meaning of NRS/NRI needs further studies.

7. Line 137 states that there is obvious variation of the MNase-X-ChIP signal across the genome (Fig. 1d). However, this would only be obvious in the context of a negative control, so that we can see what is a "mountain" and what is a "molehill". Without a robust statistical tests for NRD definition (as opposed to NRS), it is difficult to know how real these peaks are.

Response: We appreciate the reviewer for bringing up these concerns.

In Fig. 1d, the “variation” refers to the non-random variation of nucleosome density represented by H3 or H4 MNase-X-ChIP signals or MNase-X-ChIP input signal, which is different from the expected uniform coverage of nucleosome along the genome. We agree with the reviewer that a control is needed to see the magnitude of the variation, which can be achieved through the cell cycle experiment. We indeed observed that, compared with G1 and G2/M cells, nucleosome signals of S phase cells show more variation (Fig. 1h, 1i), and the pattern is similar with the bulk MNase-X-ChIP input (Fig. 1g). These results suggested that the non-random variation of nucleosome density along the genome of cell population maybe a result of DNA replication of S-phase cells in a cell population.

For Hi-C data, the command FindHiCCompartments from homer2 can be used to find compartments A and B using PC1 value as input. As the genome-wide pattern of NRI is highly similar with Hi-C PC1 value, we used FindHiCCompartments to detect nucleosome wrapping domains (NRDs) using NRI as input. We are not aware of additional statistical tests done by FindHiCCompartments for compartment detection. However, we showed data that NRDs maintain characteristic nucleosome wrapping features across multiple NRI groups (Fig. S2a), bin resolution (Fig. S2b) and histone types (Fig. 2c), suggesting that the qualification of nucleosome wrapping states by NRDs is robust.

Minor comments

8. Line 129. *It is not likely that mononucleosomes are the dominant digestion product, compared to all others as a group, which is the appropriate comparison group. Mono vs non-mono needs to be quantified.*

Response: We thank the reviewer for bringing this issue to our attention. We intended to point out that in these digestion conditions, the mono-nucleosomal DNA band is brighter than other DNA bands, such as di- or tri- nucleosomal DNA. We have rephrased the text accordingly in the revised manuscript (line 134).

9. *It is very difficult to figure out which species is being used in some experiments. Fig. 2b as one example.*

Response: We thank the reviewer for pointing out this issue. We had noted the species at the top of Fig. 2a, 2c, 2d. We also added header “Statistics of NRDs of *M. Musculus*” in Fig. 2b.

10. Line 143. *“we also need to figure out whether the variation of nucleosome density is due to experimental error of wrapping-seq or due to unsynchronized DNA replication in a cell population.” This seems like a false dichotomy. What exact “experimental error” is the reviewer referring to? How does “unsynchronized DNA replication” have anything to do with this? The data are from cell population averages, and so any potential cell cycle issue should be averaged out.*

Response: We thank the reviewer for noting this issue. We have rephrased the text as “due to potential extraction bias of chromatin or DNA during wrapping-seq or due to genome copy number variation as a result of DNA replication of the S-phase cells.”

Reviewers' comments:

Reviewer 1

I appreciate the authors addressing my concerns regarding the unwrapping and rewrapping process of DNA during MNase enzyme digestion. It is now clearer to me. However, upon reviewing the comments and responses from the second reviewer, I think about the role of MNase's A/T bias on NRI value. Surprisingly, I found that an arbitrary model involving MNase A/T bias alone can explain the conserved pattern of NRI without the role of nucleosome wrapping. Details are elaborated below:

To evaluate the role of A/T bias in the determining NRI values, I constructed an arbitrary model as below. Suppose I have a naked DNA chain of 10 Mb (with no nucleosomes bound), divided into 100 100-kb DNA bins. Each 100-kb bin is composed of 400 250-bp fragments. I can estimate the number of long fragments (a) and short fragments (b), as noted in Fig.1a of the manuscript. Before MNase digestion ($t=0$), $a=400$ and $b=0$ for every 100-kb bin. When MNase cuts the DNA into pieces, the value of a decreases while b increases. Assuming MNase digestion process is randomly for each 100-kb bin, $a(t)$ would decay exponentially with time t. Therefore, I can roughly express $a(t)$ as $400 \cdot \exp(-\lambda \cdot t)$ and $b(t)$ as $400 \cdot (1 - \exp(-\lambda \cdot t))$, where λ is a parameter related to cleavage efficiency, with each 100-kb bin having a specific λ value.

For each 100-kb bin, I first randomly assigned λ values within the range of 0.1 to 0.3. Then I calculated $a(t)$ and $b(t)$ as well as NRS and NRI for all 100-kb bins at time $t=1$. Subsequently, I recalculated the NRS and NRI for all bins at additional time points, such as $t=2, 3, 4,$ and 5 . Comparing the NRS and NRI values at different time points, I found that the NRI value for each 100-kb bin remained almost conserved, similar to the results observed in the manuscript. Considering that we only involved cleavage efficiency in the estimation, the conserved NRI values challenge the speculation that higher NRI values represent tighter DNA wrapping on nucleosomes, as well as the conclusions of the manuscript.

Based on the results from the arbitrary model illustrated above, I strongly suggest that the authors provide validated evidence to demonstrate that higher NRI values are caused by tighter nucleosome wrapping, rather than other factors such as MNase A/T bias. Otherwise, the conclusion in the manuscript will not be solid.

Response: We thank the reviewer for proposing this model. If we understand correctly, the key parameter for this model is “cleavage efficiency”, which will directly determine the production rate of shorter DNA fragments, and thus NRS/NRI. The reviewer assigned each 100-kb bin with a specific λ value, and did not assign specific DNA sequence for each bin, it probably means the “cleavage efficiency” will not change during subsequent “in silico” digestion. Thus, it is reasonable that “NRI value for each 100-kb bin remained almost conserved”. However, we did not see a correlation between these modeled results and MNase A/T bias.

Under “in tube” digestion, MNase constantly digests different DNA fragments and encounter different nucleosome conformations, thus the “lambda” for each bin should be dynamic along digestion time. Considering this point, this model would be helpful to model the contribution of DNA sequence, nucleosome conformations and other factors to NRS/NRI.

In addition, we provided supporting evidence that NRI values and NRDs are not artifacts of MNase A/T bias as described as following:

It has been reported that the genome chromatin can be evenly fragmented by Tn5 after SDS-enhanced chromatin opening at high temperature (Shanshan Ai *et al.*, 2019, Nature Cell Biology, <https://doi.org/10.1038/s41556-019-0383-5>). We have confirmed this effect. As shown by Response Fig. 1a, most of the genome chromatin was fragmented below 500 bp by Tn5 after chromatin opening. Thus, we prepare libraries from Tn5 fragmented “opened chromatin” and genome DNA. After sequencing, we found that most of the fragments within the libraries were around 150 bp (Response Fig. 1b). Consist with Shanshan Ai *et al.*, we did not observe typical peaks as in ATAC-seq, but we observed even genome coverage from the Tn5 fragmented “opened chromatin” library (Response Fig. 3, tracks of “Tn5-chromatin”).

Response Fig. 1. a. Agarose gel shows the DNA sizes after digesting the “opened chromatin” with increasing amount of Tn5. b. The distribution of inserted fragment size in libraries of Tn5 digested chromatin and Tn5 digested genomic DNA (gDNA).

It has been reported that Tn5 insertion preference is complex, and cannot be explained by either DNA motif signature or AT/GC content (Houyu Zhang *et al.*, 2021, NAR Genomics and Bioinformatics, <https://doi.org/10.1093/nargab/lqab094>). We observed complex base distribution around the insertion sites of Tn5 in chromatin and genomic DNA libraries (Response Fig. 2), which is not simply biased towards A/T base pair as MNase. When we calculate “nucleosome wrapping score (NRI)” using Tn5 digested genomic DNA, we found the pattern is roughly inverse to the MNase NRI, excluding the possibility that Tn5 insertion bias can contribute to nucleosome wrapping domain. Importantly, when we calculated NRI using Tn5 digested “opened chromatin”, we found that the resulting nucleosome wrapping pattern is similar to MNase NRI. Genome-wide analysis showed well positive correlation between MNase NRI and Tn5 chromatin NRI, but negative correlation between MNase NRI and Tn5 genomic DNA NRI. Taken together, we demonstrated that, combining with chromatin opening treatment, Tn5 can also be used to analyze nucleosome wrapping states. Moreover, we provided supporting evidence that NRI values and NRDs are not artifacts of MNase A/T bias.

Response Fig. 2, Meta profiles show the base frequency around the 5' or 3' cut site of DNA fragments from libraries of Tn5 digested chromatin and Tn5 digested genomic DNA.

Response Fig. 3, IGV tracks show the distribution of MNase NRI, Tn5-chromatin NRI and Tn5-gDNA NRI along chromosome 1 with a zoom-in view. The track of “Tn5-chromatin” shows the even coverage of Tn5 digested “opened chromatin” library.

Response Fig. 4, Dot plots show the genome-wide correlation between MNase NRI and Tn5 Chromatin NRI (left panel) or Tn5 genomic DNA NRI (right panel).

Reviewer 2

In this latest revised version of the manuscript, the authors further respond to reviewer 3's concerns that nucleosome-wrapping scores (NRI) may be the result of underlying AT-bias in MNase digestion. Reviewer 3 proposes new analysis and a potential experiment to try and assess the impact of MNase on NRI. Reviewer 3 argues that rather than plotting AT content of protected fragments (Figure 2g), the authors should plot AT content of immediately adjacent (digested) regions to better assess the correlation between AT content of digested regions and protected fragment size. However, the results of this analysis may not be useful in definitively determining the contribution of MNase bias to the authors data because of the higher incidence of AT content in heterochromatin areas with low Pol II. As the authors provide evidence that Pol II levels are strongly correlated with NRI, it would be expected that heterochromatin regions with high AT content should show low NRI, even if underlying sequence had no impact on wrapping score. Therefore, even if the analysis proposed by reviewer 3 were to show high AT content at regions of low wrapping, it would not be clear whether this is due to low transcription or MNase bias. I also agree with the authors that the experiment proposed by the reviewer 3 to digest naked DNA with MNase would not be helpful in determining the contribution of AT bias to NRI.

Response: We appreciate the reviewer's support and clarification.

REVIEWERS' COMMENTS

Reviewer #1 (Remarks to the Author):

I thank the authors for providing evidence that MNase and Tn5 chromatin NRI values share similar profiles, which successfully addresses my concern that the MNase NRI might result from the A/T bias of MNase rather than reflecting chromatin characteristics. While the statement that 'higher NRI values are caused by tighter nucleosome wrapping' seems physically plausible, I remain concerned that other factors may contribute to the NRI values beyond the strength of nucleosome wrapping. Although I lack evidence to support this concern, and I understand it may be difficult to fully address in this manuscript, I hope future work will provide robust evidence to substantiate the claim that 'higher NRI values are caused by tighter nucleosome wrapping,' rather than relying on physically reasonable speculation. Lastly, I appreciate the authors' patience and effort in addressing my concerns throughout the review process.

Response: We appreciate the reviewer for highlighting this crucial point. We agree that “other factors may contribute to the NRI values beyond the strength of nucleosome wrapping”. We rephrased the following statement in the discussion part to remind the readers of the limitation of this study: “However, due to the limitation of wrapping-seq, only one dimension of the physical wrapping states of nucleosomes *in vivo* can be deduced. Moreover, other factors beyond the strength of nucleosome wrapping may contribute to the NRS or NRI values. Other techniques, such as single-molecule and structural techniques, are needed to depict the biochemical basis of nucleosome wrapping states *in vivo*.”

The authors used the MNase-X-ChIP-seq protocol to map the lengths of DNA fragments resulting from MNase enzyme cleavage across genome-wide. Utilizing this data, they introduced metrics, namely the Nucleosome Wrapping Score (NRS) and Nucleosome Wrapping Index (NRI), to describe the average wrapping states of nucleosomes within specific genomic regions. Upon analyzing the distribution of nucleosome wrapping states across genome-wide, the authors identified a very intriguing phenomenon: the genome can be partitioned into tightly wrapped nucleosome wrapping domains (TiNRDs) and loosely wrapped nucleosome wrapping domains (LoNRDs); TiNRDs and LoNRDs precisely coincide with A compartment domains and B compartment domains, respectively. I have the following concerns:

Major

1. The research community dedicated to investigating 3D chromatin organization broadly agrees that chromosomes are hierarchically organized into large A/B compartments, further divided into smaller topologically associating domains (TADs). The work here presented, claims to reveal a novel level of chromatin organization. However, it simultaneously emphasizes that TiNRDs and LoNRDs precisely coincide with widely studied A and B compartments, respectively. Thus, the clarity of the novel level of chromatin organization is needed.
2. The work here presented, also claims to reveal a novel principle of chromatin organization. However, the investigation primarily focuses on the wrapping length of DNA on nucleosomes, uncovering a novel phenomenon or feature within euchromatin and heterochromatin - specifically, that the wrapping lengths of DNA on nucleosomes within euchromatin are likely longer than those within heterochromatin. Despite this revelation, the study does not establish a causal relationship between this feature and the formation of NRDs or compartments. Further clarification is necessary to elucidate the proposed novel principle of chromatin organization.
3. Fig. 3a displays a clear positive correlation between H3 NRI(140) and DNase I sensitivity, while Fig. 1d reveals a similar correlation between H3 NRI(140) and H3 ChIP signals. This indicates that chromatin regions with high DNase I sensitivity (indicating low nucleosome density) also exhibit high H3 ChIP signals (indicating high nucleosome density). To enhance the study's credibility, the author should provide an explanation for this apparent contradiction.
4. In the section titled 'Transcription promotes nascent nucleosome wrapping,' the author labels nascent nucleosomes using EdU and subsequently conducts a wrapping-seq experiment to explore nucleosome wrapping dynamics during DNA replication. In this section, it is crucial

for the author to initially rule out the influence of EdU labeling on the distribution of fragment length (as shown in Fig. 1a) and the calculation of NRI/NRS values. Without such exclusion, it becomes challenging to determine whether the observed differences in NRS values are attributed to DNA replication or EdU labeling.

5. In Fig. 1e, certain chromatin regions exhibit a notable change in NRS under different digestion time, while NRI remains relatively constant. Consequently, I recommend that the author uses NRI instead of NRS as a metric for assessing nucleosome wrapping states in the section 'Transcription promotes nascent nucleosome wrapping' and in Fig. 4. Considering the author's claim that 'Input' (parental) and 'Pulldown' (nascent) nucleosomes in each 'Nascent' or 'Chase' condition undergo the same MNase digestion, it is expected that NRI values would yield similar results.
6. In Fig. 4d, the author employs a bin size of 100kb for the NRS calculation, which is close to the size of DNA replication domains. It's worth noting that chromatin is typically partitioned into TADs with a median size of 185kb (as reported in Cell 159, 1665-1680), and TADs are recognized as stable units of replication timing regulation (as reported in Nature 515, 402-405, 2014). Given this context, I recommend that the bin size used to measure nucleosome wrapping states during the DNA replication process be much smaller than the size of replication domains.

Minor

7. As shown in Figure S1e, the NRI values vary with respect to the size of each bin when computing the NRS values. It is necessary to specify the bin sizes of NRI values used in other figures in this work.
8. Does the "MNase input" in Fig. 1d mean classical MNase-seq data or MNase-seq data from wrapping-seq? Please use some terms to distinguish them.
9. One type error in line 175 "Correspondingly, quantification of NRS in these peak regions showed that nucleosomes in H3K4me1 or H3K4me3 peaks regions have higher H3 NRS(140) **that**(should be than) those in H3K9me3 or H3K27me3 peaks regions (Fig. S3a)."